# Correlation of membrane protein conformational and functional dynamics

Raghavendar Reddy Sanganna Gari[1,2], Joel José Montalvo-Acosta[3], George R. Heath [1,4], Yining Jiang[2], Xiaolong Gao [1], Crina M. Nimigean [1,2], Christophe Chipot [3,5✉] & Simon Scheuring [1,2✉]

Conformational changes in ion channels lead to gating of an ion-conductive pore. Ion flux has been measured with high temporal resolution by single-channel electrophysiology for decades. However, correlation between functional and conformational dynamics remained difficult, lacking experimental techniques to monitor sub-millisecond conformational changes. Here, we use the outer membrane protein G (OmpG) as a model system where loop-6 opens and closes the β-barrel pore like a lid in a pH-dependent manner. Functionally, single-channel electrophysiology shows that while closed states are favored at acidic pH and open states are favored at physiological pH, both states coexist and rapidly interchange in all conditions. Using HS-AFM height spectroscopy (HS-AFM-HS), we monitor sub-millisecond loop-6 conformational dynamics, and compare them to the functional dynamics from single-channel recordings, while MD simulations provide atomistic details and energy landscapes of the pH-dependent loop-6 fluctuations. HS-AFM-HS offers new opportunities to analyze conformational dynamics at timescales of domain and loop fluctuations.

[1] Weill Cornell Medicine, Department of Anesthesiology, New York, NY, USA. [2] Weill Cornell Medicine, Department of Physiology and Biophysics, New York, NY, USA. [3] Laboratoire International Associé CNRS and University of Illinois at Urbana-Champaign, Université de Lorraine, Vandœuvre-lès-Nancy, France. [4] Present address: Astbury Centre for Structural Molecular Biology, School of Physics & Astronomy, University of Leeds, Leeds, UK. [5] Present address: Department of Physics, University of Illinois at Urbana-Champaign, Urbana, IL, USA. ✉email: chipot@illinois.edu; sis2019@med.cornell.edu

Membrane proteins (MPs) reside in the plasma membrane and perform various biological processes including ion transport, substrate transport, and signal transduction. Function-related conformational changes in MPs occur on time scales ranging from nanoseconds to seconds[1–3]. Recent advances in structural biology, notably in single-particle cryo-electron microscopy (cryo-EM), have enabled the solving of many MP structures. However, obtaining time-resolved dynamic information of MPs in their membrane environment is still a major challenge.

X-ray and cryo-EM structures can report several conformations of MPs. However, they do not report structural dynamics, rates of conformational conversions, equilibrium distributions of states, and correspondence with functional states[4]. Several techniques have paved the way for the analysis of protein dynamics: nuclear magnetic resonance (NMR) provides information about protein dynamics, but is often restricted to small MPs, and requires difficult and costly isotope labeling[5]. Double electron-electron resonance (DEER) spectroscopy relies on the insertion of probes of considerable size into the MP[6]. Fluorescence resonance energy transfer (FRET) reports about conformational dynamics, but necessitates the insertion of fluorescence probes at structurally predetermined sites[7]. Taken together, NMR, DEER, and FRET provide structural dynamics of molecules, but rather indirectly and require preexisting structures to place probes and interpret the data.

High-speed atomic force microscopy (HS-AFM[8]) images label-free samples (DNA, soluble proteins, MPs, and intrinsically disordered proteins[9–11]) at ~1 nm lateral, ~0.1 nm vertical, and ~100 ms temporal resolution, in aqueous environment and at ambient temperature and pressure. HS-AFM has the additional advantage that the environment, such as the buffer composition[12,13], light[14], temperature[15], and force[16,17], can be altered during movie acquisition. However, 100-ms temporal resolution is too slow to characterize many dynamic biological processes including loop dynamics, domain motions, and fast enzymatic processes that occur on microsecond-to-milliseconds timescales[3]. To overcome this limitation, we previously introduced HS-AFM height spectroscopy (HS-AFM-HS), where the HS-AFM probe is placed at a fixed x, y-position on the sample, and the height fluctuations in the z-direction under the AFM probe are monitored with angstrom precision and 10-μs temporal resolution[18]. Therefore, we expected that this approach was equally powerful to assess the conformational dynamics of integral MPs.

Thus, here we use HS-AFM-HS to characterize the microsecond timescale conformational changes of an integral-MP model system, i.e., the outer membrane protein G (OmpG) in a membrane environment. OmpG is imbued with both structural and functional simplicity, where the motions of a single loop are supposed to directly open and close the ion conducting pore, transitions which can be characterized with single-channel recordings. OmpG has been extensively studied, both structurally and functionally, owing to its promising applications in biosensing[19–23]. OmpG is a 14-stranded β-barrel with 7 extracellular loops[20,23]. X-ray structures of OmpG revealed that at neutral pH, the channel (barrel) is open, while at pH 5.6 access to the barrel is blocked by loop-6, which folds into the barrel lumen in a lid-like manner[23]. Contact mode AFM imaging confirmed the open and closed states of OmpG in lipid bilayers, at different pH values[24]. Solution and solid-state NMR of OmpG in DPC micelles[20] and in lipid bilayers[25], respectively, revealed that the extracellular loops of OmpG were highly dynamic at acidic pH. NMR detected three conformation ensembles resembling the open and closed states observed in the X-ray structures, and an intermediate conformation. This work also showed that loop-2

and loop-6 could move in a correlated manner[26]. Single-channel recordings showed that at neutral pH, OmpG was mostly open, but exhibited flickers to a closed state[19,21,22]. At pH 5.0 single-channel recordings revealed both open and closed states, though the closed state was the higher probability state[19,21,22]. Deletion of loop-6 together with D215, lead to open probabilities >95% independent of the pH[22]. Silencing the motion of loop-6 by tethering it to another neighboring strand also lead to an open probability of ~99% independent of the pH[21]. In addition, pinning of loop-6 to the membrane lead to an open probability of ~99%[27]. Therefore, the fluctuations of loop-6 are responsible for gating. Thus, the structural characterizations by X-ray crystallography[23] and conventional AFM[24] provided static snapshots of the most probable state under a given condition, while NMR reported loop flexibility[20,25], and electrophysiology showed that open and closed states coexisted in all conditions[19,21,22]. However, a direct correlation between structural conformations and channel functional states was not possible because of the lack of a time-resolved single-molecule structural technique to cover timescales similar to those of electrophysiology measurements.

HS-AFM-HS was used to directly monitor conformational dynamics of loop-6 in membrane-reconstituted OmpG at submillisecond temporal resolution. We correlate HS-AFM-HS conformational gating dynamics of loop-6 with single-channel recordings that provide kinetic information about channel gating. We use molecular dynamics (MD) simulations to analyze pH-dependent gating dynamics and construct free-energy landscapes of OmpG. This multi-pronged analysis allows us to establish the structure-dynamics-function relationship of a simple membrane protein. We anticipate that such correlation studies will offer new opportunities in membrane biology to better understand membrane proteins at work at high spatiotemporal resolution.

## Results

**HS-AFM imaging of OmpG in lipid bilayers.** We reconstituted OmpG into POPE/POPG (80:20) lipids at lipid-to-protein ratios (LPR) between 0.5 and 0.7 (w:w; in our case to 0.65 mM lipid/ 0.03 mM protein and 0.92 mM lipid/0.03 mM protein). Reconstitutions resulted in 2D-crystals, where OmpG molecules packed into rows of dimers, occasionally interspersed by monomer rows[24]. OmpG protruded ~1.25 nm above the membrane on the extracellular face (Supplementary Fig. 1), in good agreement with estimates from the X-ray structure[23]. All of our data focus on the extracellular side of the channel, where the gating loop-6 is located. Time-lapse HS-AFM imaging at 5 frames per second of OmpG at pH 7.6 showed molecules predominantly in the open conformation (Fig. 1a, b), where the entrance into the β-barrel cavity is visible, albeit occasionally fluctuations of loop-6 over the pore are detected (Supplementary Movie 1, left). The correlation average (Fig. 1b, c) showed good agreement with the molecular outline of the open state X-ray structure (Fig. 1e): Loop-6 was the loop that protruded strongest in height along the β-barrel circumference (asterisk in Fig. 1c and label in Fig. 1e). In addition, loop-2 reaches outwards from the β-barrel in the X-ray structure, which is well contoured by HS-AFM (L2 in Fig. 1e, compare to Fig. 1c). The following elements guided our assignment of the HS-AFM topography with respect to the X-ray structure: the OmpG barrel (open state, Fig. 1c) displayed high protrusions on one side and low protrusions on the other side of the barrel. In the X-ray structure (open state, Fig. 1e) low protruding features (loop-7, loop-1, and loop-2) are grouped on one side of the barrel, thus, this global assessment basically constrained the barrel placement unambiguously. In addition, this placement, sets loop-2 precisely at the location where HS-AFM detects a laterally

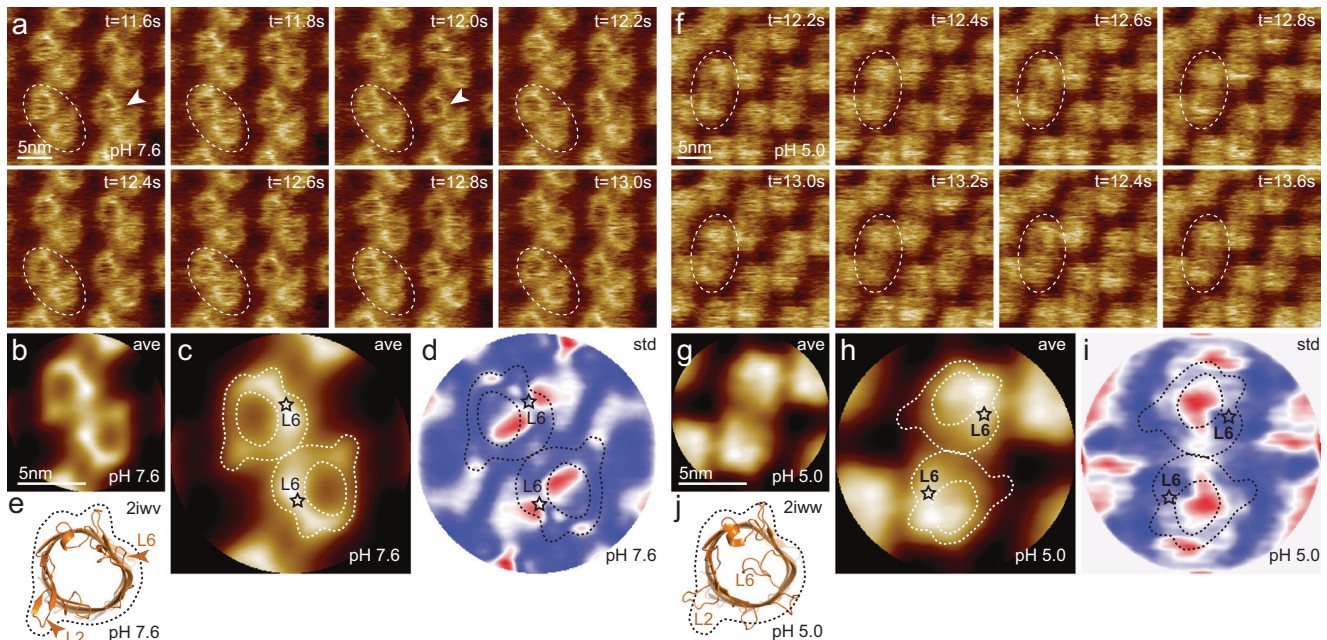

**Fig. 1 HS-AFM imaging of OmpG in lipid bilayers at pH 7.6 and pH 5.0. a** OmpG at pH 7.6 (Supplementary movie 1, left; frame rate: 200 ms per frame). A OmpG dimer is highlighted with dashed outline in all frames. Arrowheads in t = 11.6 s: Loop-6 fluctuating over the lumen. Arrowhead in t = 12.0 s: Fully open state. **b** Correlation average (n = 2752) of the HS-AFM movie frames (344 frames recorded over 68.8 s, full color scale: 0.0 nm < height < 1.25 nm, where the membrane level was set to 0.0 nm). **c** Correlation average of OmpG dimers. The topography outline (based on the molecular structure in 1e), serves as a visual guide to locate loop-6 and loop-2 in the topography and is highlighted by the dashed outline (the position of loop-6 is indicated by the asterisk based on its location in the structure (**e**)). Inner dashed outline show barrel lumen. **d** Standard deviation (std) map (n = 2752) from the averaging process in (**b**) (full color scale from blue to red: 0.05 nm < std < 0.19 nm) and topography outlines as in (**c**). **e** X-ray structure (PDB 2iwv) of the open OmpG conformation. Loop-6 (arrowhead L6) stands out of the image plane towards the viewer. Loop-2 (L2) forms a beta strand pointing away from the β-barrel, well detected by HS-AFM in the open state (**b**). **f** OmpG at pH 5.0 (Supplementary movie 1, right; frame rate: 200 ms per frame). A OmpG dimer is highlighted with dashed outline in all frames. **g** Correlation average (n = 2472) of the HS-AFM movie frames (309 frames recorded over 61.8 s, full color scale: 0.0 nm < height < 0.7 nm, where the membrane level was set to 0.0 nm). **h** Correlation average of OmpG dimers. For comparison, the topography outline of the open state (**e**) is shown (the position of loop-6 is indicated by the asterisk). **i** Standard deviation (std) map (n = 2472) from the averaging process in (**g**) (full color scale from blue to red: 0.04 nm < std < 0.07 nm) and topography outlines as in (**h**). **j** X-ray structure (PDB 2iww) of the closed OmpG conformation shown in the same orientation as in (**e**). Loop-6 (L6) folds over the β-barrel lumen in a lid-like manner. Loop-2 (L2) does not form a β-strand in the closed state, in agreement with absence of topography in this region in (**h**). Black dashed line: outline based on (**e**) for comparison.

protruding nose, and indeed loop-2 forms a laterally extending β-strands in the X-ray structure. Finally, this placement, also sets loop-6 to the location where HS-AFM detects the highest protrusion, which in addition is the region that showed largest mobility in the standard deviation (std) map (Fig. 1d–i). The solid state NMR structure of OmpG revealed, however, that the β-barrel structure extended into loop-3 and loop-4, where loop-4 was the most stable[25], which might suggest a different structural alignment. We note that loop-4 is still in the high protrusion area of our topography, however, loop-3 might be flexible and not give a strong average topography signal. Concomitant with calculation of the correlation average, a standard deviation map highlighting molecular flexibility was calculated[28], which reports that loop-6 swings from its average position mainly over the β-barrel but also occasionally outwards away from the β-barrel (Fig. 1d, red-shaded areas). At pH 5.0 the channel is generally in the closed state (Fig. 1f, g): The pore is occluded in the correlation average (Fig. 1g, h) in agreement with loop-6 covering the β-barrel in the closed state X-ray structure (Fig. 1j). This conformational change is accompanied by an overall decrease of the protrusion height by ~0.5 nm. Interestingly, the overall molecular shape is also different when compared to the open state, notably the peripheral lobe occupied by loop-2 in the open state is not detected, which is understandable in light of the X-ray structure that shows that loop-2 does not form a β-strand in the closed state and is in a different orientation (L2 in Fig. 1j, compare to Fig. 1h, and also to

the open state X-ray structure Fig. 1e). Finally, the standard deviation map shows that loop-6 covering the pore is highly flexible (Fig. 1i), likely a signature of occasional fluctuations to the open state (Supplementary Movie 1, right). Further regions with increased mobility at acidic pH correspond to the locations of loop-4 and to a minor extent of loop-2. Thus, HS-AFM corroborates the open and closed conformations of OmpG at neutral and acidic pH, respectively, consistent with the X-ray structures[23], and NMR conformation ensembles resembling open and closed states[26]. Furthermore, we see different loop-2 average configurations at neutral and acidic pH, which might be an indirect and static signature of the correlated loop-6 / loop-2 action as observed by NMR[26]. While the standard deviation maps comprise signatures of loop-6 fluctuations, HS-AFM imaging cannot be correlated with the functional dynamics[19,21,22], because its image acquisition speed is too slow.

**Correlation between ion flux and loop-6 dynamics of OmpG in lipid bilayers.** To compare OmpG structural and functional dynamics, we recorded single-channel currents at pH 7.6 and pH 5.0. Refolded OmpG (in βOG) was added at nanomolar concentration to a preformed free-standing lipid bilayer, which separated two aqueous chambers in an electrophysiology setup (see Methods). Following insertion of a single OmpG molecule into the membrane, a voltage across the membrane was applied,

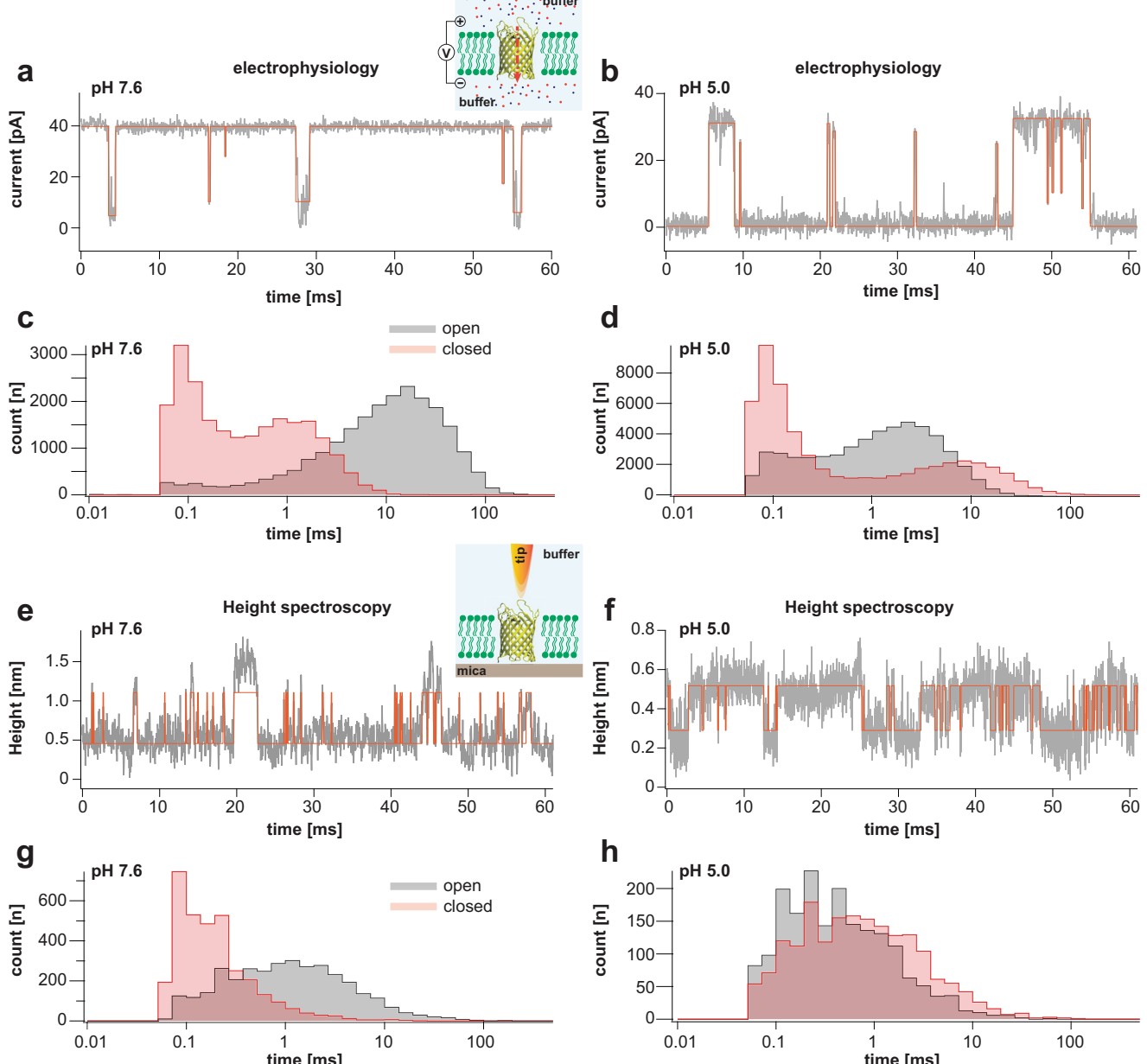

**Fig. 2 Single channel electrophysiology and HS-AFM height spectroscopy recordings of OmpG in lipid bilayers.** Representative 60-ms segments of OmpG single channel recordings at pH 7.6 (**a**) and pH 5.0 (**b**) at +40 mV membrane potential (longer traces in Supplementary Figs. 2 and 3). Cartoon representation of single channel recording experimental setup is shown in inset of (**a**). OmpG (yellow) in open state (PDB:2IWV) is placed in a lipid bilayer (green) surrounded by buffer (light blue shade) and potassium and chloride ions are shown as red and blue spheres. Red arrow indicates ion flow through OmpG in response to voltage application. Dwell time histograms of open and closed states at pH 7.6 (**c**) and pH 5.0 (**d**) from single-channel recordings (see Supplementary Table 1). Representative 60-ms segments of OmpG HS-AFM-HS recordings at pH 7.6 (**e**) and pH 5.0 (**f**) (longer traces in Supplementary Fig. 4). Cartoon representation of HS-AFM height spectroscopy experimental setup is shown in inset of (**e**). An oscillating AFM tip (orange) detects conformational changes of loop motion. Dwell time histograms of open and closed states at pH 7.6 (**g**) and pH 5.0 (**h**) from HS-AFM-HS recordings (Supplementary Table 2). In HS-AFM-HS the low state represents the open state, where the HS-AFM tip can descend into the β-barrel, and the high state represents the closed state, where loop-6 covers the beta barrel barring access of the HS-AFM tip to the cavity. All current-time and height-time traces were filtered at 20 kHz during analysis. The state dwell-time histograms are shown using log binning for better visualization of the components[49]. Red traces in (**a**) and (**b**) represent idealized current-time traces using clampfit software. Red traces in (**e**) and (**f**) represent idealized height-time traces using the STaSI algorithm (see Methods).

and the resulting ionic current was measured. Our single-channel recordings, similar to previously reported measurements[19,21,22], revealed a mostly open channel, with a ~40pA single-channel amplitude with occasional flickers to the closed level at pH 7.6 (Fig. 2a), whereas at pH 5.0 the channel was mostly closed, with intermittent short openings (Fig. 2b). The OmpG conductance is

~1nS, indicating that the OmpG beta barrel permits high ionic throughput, in good agreement with previous studies[19]. Single-channel recordings from several different bilayer experiments at both neutral and acidic pH (Supplementary Table 1) resulted in open probabilities (Po) $Po_{(pH7.6)} = 0.95 \pm 0.01$ and $Po_{(pH5.0)} = 0.38 \pm 0.07$, respectively. At pH 7.6, fitting of the open dwell-time

distribution resulted in one exponential component with a time constant of $11.5 \pm 0.04$ ms, while the closed dwell-time distribution was fit with two exponential components: one with a time constant of $0.8 \pm 0.1$ ms, and a second that contains mainly short flickers (<100 µs) (Fig. 2c). At pH 5.0 the open component had a time constant of $1.9 \pm 0.09$ ms and the closed component $6.3 \pm 0.24$ ms (with a second component containing flickers of <100 µs, similar to those found at pH 7.6) (Fig. 2d). Distributions with multiple exponential components (Fig. 2c, d) are quite usual for ion channels, where the components represent different functional states that correspond to distinct protein conformations involved in gating. The very short flickers to the closed level encountered in both conditions are believed to correspond to ion flow interruptions due to amino acid fluctuations in the pore rather than to large protein conformational changes.

The single-channel electrophysiology experiments were performed at a 10-µs temporal resolution and reported dwell-times in the millisecond time range. Therefore, the HS-AFM movies recorded at 200 ms (Fig. 1) cannot be directly compared and related with the functional data. In order to measure dynamics at a temporal resolution similar to that of electrophysiology, we resorted to HS-AFM-HS. In this method, the oscillating AFM probe is positioned above the pore region of single OmpG molecules, and the structural fluctuations in the z-direction are measured at a ~10-µs temporal resolution. The positioning of the tip is guided by HS-AFM imaging immediately before HS-AFM-HS operation. In addition, we took advantage of the dense packing of OmpG molecules: in 2D-crystals, there is no lateral diffusion of the molecules, and in case a molecule is lost during HS-AFM-HS data capture due to instrumental lateral drift, another neighboring molecule is soon to be detected. Since OmpG gating is mainly controlled by loop-6, measurable changes in the z-direction occur when the protein undergoes conformational changes from the open to the closed state, i.e., when the tip can descend into the β-barrel cavity, or not. HS-AFM-HS measurements at the OmpG pore entrance show transitions between two distinct z-positions, with a height difference of ~5 Å (Fig. 2e, f). It is worth noting that when comparing HS-AFM-HS with electrophysiology traces (Fig. 2a, b), the overall appearance is inverted, because the open state is manifested by low height (tip entering the pore) and current with a non-zero value (ions passing through the pore), and vice versa in the closed state, where the topography is high (the tip cannot enter the pore), while the current is zero. Thus, in HS-AFM-HS, the high state corresponds to a conformation corresponding to loop-6 folding over the β-barrel (closed), and the low state corresponds to a conformation, whereby loop-6 is extended and the tip can descend into the pore (open). Expectedly, HS-AFM-HS height-time traces appear noisier than electrophysiology current-time traces, because the tip also detects intermediate states of L6 in between the functionally open and closed states. We note that at pH 5.0 (Fig. 2f), height differences between open and closed states are sometimes lower than the expected ~5 Å because the pore has high probability for closed state making it harder for the probe to penetrate deeper. In general, tip radius influences the full amplitude of height change that can be measured. Despite smaller changes in height, kinetics should not be influenced by the HS-AFM probe (see Methods).

We collected height spectroscopy from several different channels at both physiological and acidic pH (Supplementary Table 2), and determined state probabilities at pH 7.6, $P_{L(pH7.6)} = 0.9 \pm 0.05$, and at pH 5.0, $P_{L(pH5.0)} = 0.33 \pm 0.09$. $P_L$ represents open probability from height spectroscopy, where L is the low height, i.e. open, state. These probabilities are in good agreement with the $Po$ values from electrophysiology data (Supplementary Table 1). Dwell-time analysis of the low (open) and high (closed) states in the HS-AFM-HS traces (see Methods[29]) revealed time constants of $2.16 \pm 0.12$ ms and $0.23 \pm 0.2$ ms, respectively, at pH 7.6 (Fig. 2g). At pH 5.0, the time constants of low (open) and high (closed) states were $0.77 \pm 0.15$ ms and $1.52 \pm 0.14$ ms, respectively (Fig. 2h). Thus, while the state probabilities $Po_{(pH7.6)} = 0.95 \pm 0.01$ and $Po_{(pH5.0)} = 0.38 \pm 0.07$ in electrophysiology and $P_{L(pH7.6)} = 0.9 \pm 0.05$, and $P_{L(pH5.0)} = 0.33 \pm 0.09$ in HS-AFM-HS are very similar, the kinetics were somewhat different, i.e., the open state time constant at neutral pH was ~5 times longer in electrophysiology than in HS-AFM-HS, and the time constants of the remaining conformations (closed state at neutral pH and both open and closed states at acidic pH) are ~2–4 times longer in electrophysiology experiments. Also, in single channel electrophysiology experiments, the closed state at both pHs has bimodal dwell-time distributions (Fig. 2c, d), whereas in HS-AFM-HS we find only one distribution (Fig. 2g, h). We attribute these differences to the different measurables of the two techniques: Electrophysiology detects longer ion flow interruptions caused by loop-6 dynamics and short interruptions due to amino acid fluctuations in the pore, while HS-AFM-HS senses the conformational dynamics of loop-6. The shortening of the state dwell times in HS-AFM-HS with respect to electrophysiology suggests that loop-6 structural transitions occur faster than the corresponding functional state transitions, an observation that may well be explained by loop-6 temporarily hovering back and forth above the β-barrel, but not entirely occluding the ion-conduction pathway. In line with our observations and interpretation, prior NMR studies showed that loop-6 (and other loops) were highly dynamic[26,27]. The NMR relaxation dispersion data revealed that residues in loop-6 moved on sub-millisecond time scales[26], which is in good agreement with the time scales of loop-6 motions observed by HS-AFM-HS.

**Structural dynamics of OmpG in lipid bilayers from molecular dynamics simulations.** Having assessed OmpG gating in terms of functional kinetics using single channel electrophysiology and conformational kinetics using HS-AFM-HS, we aimed at gaining atomic scale structural insights about how loop dynamics and interactions mediate gating using molecular dynamics simulations (MDS)[30] (Supplementary Fig. 5). In particular, loop-6 is the primary region responsible for the open/close transitions undergoing the largest pH-dependent conformational change, covering the stable β-barrel (Supplementary Fig. 6) in a lid-like manner under acidic conditions. To this end, we performed all-atom MDS of OmpG in a lipid bilayer environment, using the two available crystal structures for the open (at 2.3 Å resolution) and closed (at 2.7 Å resolution) states at pH 7.5 and pH 5.6, respectively[23]. In these structures, loop-6 is 11-residue long (218–228) in the open state, and 18-residue long (217–234) in the closed state, because several residues that were part of β-strand-12 in the open state participate in loop-6 in the closed state. Starting with these structures, we performed unbiased 3µs-long MDS and found that in the OmpG open state at physiological pH loop-6 exhibits a high mobility in the solvent (Supplementary Movie 2, 3), while in the closed state at acidic pH loop-6 is twisted on top of the β-barrel and partially confined into the lumen (Supplementary Movie 4, 5).

**Analysis of OmpG gating from atomistic simulations.** To characterize the conformational fluctuations of loop-6 at physiological and acidic pH, we first defined a geometrical observable, ζ, the Euclidean distance between the $C_\alpha$ atoms of residues R228 in loop-6 and E52 at the top of β-sheet S3 on the opposite side of the barrel (Fig. 3a, top). The $C_\alpha$ of R228 is highly mobile, whereas the mobility of $C_\alpha$ of E52 is nearly nil (Supplementary Fig. 7). The

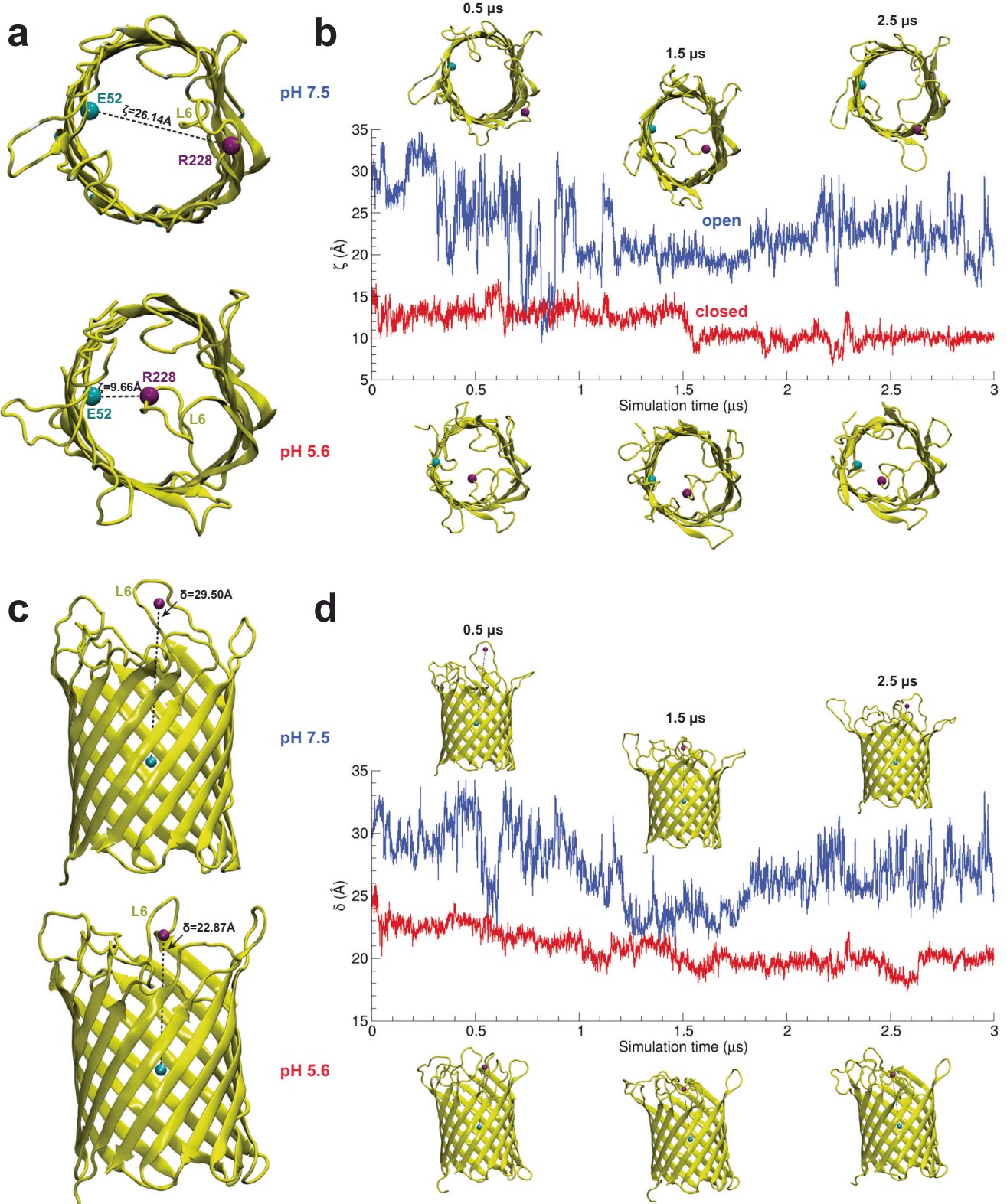

**Fig. 3 pH-dependent gating analysis of OmpG using molecular dynamics simulations. a** Collective variable $\zeta$ is the distance between the $C_\alpha$ atoms of residues E52 (cyan spheres) and R228 (purple spheres). $\zeta$ in the open and closed state crystal structures are 26.1 Å and 9.6 Å, respectively. **b** Time series of $\zeta$ for the open and closed states of OmpG. Representative structures in the open (top) and closed (bottom) states from simulations at 0.5, 1.5, and 2.5 μs are shown (Supplementary Movie 2, 3). **c** collective variable $\delta$ between the centers of mass (COM) of the backbone atoms for the barrel (cyan balls) and the $C_\alpha$ atoms for residues 220–228 of loop-6 (purple spheres). Corresponding $\delta$ values in X-ray structures of OmpG in the open and closed states are indicated. **d** Time series of the $\delta$ for the open and closed states of OmpG. Representative structures of OmpG in the open (top) and closed (bottom) states at 0.5, 1.5, and 2.5 μs are shown (Supplementary Movie 4, 5).

two residues interact strongly in the closed state (Fig. 3a, bottom). $\zeta$, therefore, is an excellent metric to evaluate loop-6 fluctuations in the x,y-plane, because (i) it measures the displacements of L6 relative to an almost fixed point, and (ii) it is characteristic to the end- (and intermediate-) states in the OmpG gating process.

The time-evolution of $\zeta$ in the OmpG open state (Fig. 3b, blue trace) shows how loop-6 was elongated and occasionally protruded beyond the edge of the barrel during the first 1.2 µs, but mainly fluctuated around the conformation captured in the X-ray structure, with an average $\zeta$ of 24.5 Å. During this period, loop-6 fluctuations were maximum, with a standard deviation of 5.3 Å. The MDS analysis of the loop-6 qualitatively corroborate the HS-AFM imaging analysis at neutral pH: The standard deviation map concomitantly calculated during particle averaging and formerly shown to allow mapping flexible protein domains[28], documents bilobed loop-6 fluctuations to both sides of the barrel (Figs. 1d, 3b–d, blue traces). Between 1.2 µs and 2.1 µs, loop-6 interacts with loop-1, thereby diminishing its fluctuations. Viewed from the top, the β-barrel appears partially closed (inset), but this state is likely fully conductive. This is in qualitative agreement with our findings where the low (open-pore) state measured with HS-AFM-HS was shorter-lived (and intersected by short apparent closures) than the open state from single channel recording (compare Fig. 2e with Fig. 2a). During this period, loop-6 had lowest mean ± std $\zeta$ of 20.4 ± 1.4 Å. During the last 0.9 µs of the simulation, loop-6 regained some flexibility, fluctuating between interactions with both loop-1 and loop-4. During this period, a fully open structure is observed again, with mean ± std $\zeta$ of 22.7 ± 2.5 Å.

In contrast, the OmpG closed state (Fig. 3b, red trace) is characterized by a distorted and rigid conformation of loop-6. At low pH, $\zeta$ follows, indeed, a bimodal distribution. During the first 1.5 µs of the simulation, loop-6 interacted with loop-7, with mean ± std $\zeta$ of 13.0 ± 1.0 Å, while during the latter 1.5 µs, loop-6 interacted with loop-1 and loop-2, with mean ± std $\zeta$ of 9.9 ± 0.8 Å. Thus, at neutral pH loop-6 always displayed more dynamics and larger separation distance to E52 than in the closed state in acidic conditions. The difference between $\zeta$ values of the open and closed states is 16.5 Å in the X-ray structures and up to ~25 Å in MDS, representative of the loop-6 lid-like movement towards the opposite side of the barrel.

A second metric, $\delta$, the Euclidean distance between the centers of mass of the backbone atoms of the barrel and the $C_\alpha$ atoms of loop-6 (Fig. 3c), was used to capture the fluctuations of loop-6 with respect to the $z$–axis of the barrel. The OmpG barrel is a very rigid structure, as evidenced by low RMSD mean values of 1.3 ± 0.1 Å and 1.2 ± 0.1 Å for the open and closed states, respectively, from unbiased simulations (Supplementary Fig. 6). These RMSD values are comparable with values from solution and solid-state NMR of 1.43 ± 0.30 Å and 2.06 ± 0.42 Å, respectively[20,23]. Thus, computing the time-evolution of $\delta$ from the 3 µs MDS of the OmpG open (Fig. 3d, blue trace) and closed (Fig. 3d, red trace) states reported loop-6 fluctuations along the $z$–axis. The analysis yielded mean ± std $\delta$ values of 29.0 ± 2.2 Å, 24.4 ± 1.6 Å and 26.9 ± 1.7 Å, respectively, for the three time-regimes outlined above (0.0 µs–1.2 µs, 1.2 µs–2.1 µs, and 2.1 µs–3.0 µs) during the open state simulation. In the closed state, loop-6 is bent far down the $z$–axis, with mean ± std $\delta$ values of 21.7 ± 1.1 Å and 19.6 ± 0.6 Å in the first and second halves of the simulation. Thus loop-6 moves on average by ~6.1 Å downwards to close the β-barrel.

In summary, MDS show that loop-6 is laterally and vertically mobile at neutral pH, while it is conformationally constrained at acidic pH. These atomistic details are to some extent reflected in the HS-AFM standard deviation maps where loop-6 displays large fluctuations to both sides of the β-barrel at pH 7.6, while it is confined to the barrel pore in the closed state.

**Free-energy landscape of OmpG gating**. Next, we attempted two additional all atom MDS, both 3 µs long, with OmpG in open and closed states, yet in the opposite pH condition. Starting from the open conformation crystal structure, we assigned the protonation states characteristic of pH 5.6, i.e., the closed conformation, and symmetrically, starting with the closed state crystal structure with the protonation states for a neutral pH 7.5 (Supplementary Table 3). Our goal was to assess whether or not a spontaneous state transition could be observed by simply swapping the native protonation states between conformations. The statistics were the following: open-to-close, $\zeta_{(start)}$ 26.14 Å to $\zeta_{(end)}$ ~26 Å and $\delta_{(start)}$ 29.50 Å to $\delta_{(end)}$ ~30 Å, and closed-to-open $\zeta_{(start)}$ 9.66 Å to $\zeta_{(end)}$ 18.5 Å and $\delta_{(start)}$ 22.87 Å $\delta_{(end)}$ ~27 Å (Supplementary Fig. 9). Thus, only during the closed-to-open transition 3 µs MDS a minor conformational change was observed. The closed-to-open transition at pH 7.5 appears to be facilitated by unprotonated residues E15, E17 and E52 that induce electrostatic repulsion on the unprotonated acidic residues D221, D224, D225, E227, and E229 in loop-6, thereby pushing it out of the lumen. However, evidently, appreciably longer MDSs would be needed to observe a full gating transition.

Last, in order to establish the conformational equilibrium of OmpG at a given pH and associate the free-energy landscape with the gating mechanism, we turned to an advanced MDS sampling technique. The potentials of mean force (PMFs) characterizing the transitions between the closed and the open states at pH 5.6 and at pH 7.5 were determined using a multiple-walker[31] variant of the well-tempered metadynamics extended adaptive biasing force (WTM-eABF) algorithm[32], and a collective variable, $\xi$, defined as the difference between the RMSD of the current conformation with respect to the other state (closed-to-open and open-to-closed). The PMFs were obtained with four individual walkers over an aggregate simulation time of 4.8 µs and 6.8 µs, for the low- and the high-pH transition, respectively (Fig. 4). Not unexpectedly, at pH 5.6, the free-energy profile features a global minimum at $\xi = -2.3$ Å, corresponding to the closed state. In addition, a metastable state is found at $\xi = 0.3$ Å, consistent with a nearly open conformation of OmpG. In this conformation loop-6 is partially extended and bends outward of the lumen, thus likely a conductive state. It is about 4 kcal/mol higher in free energy with respect to the closed state. A family of conformations representative of the open state can be found around $\xi = 2.0$ Å, with loop-6 completely extended and oriented perpendicularly with respect to the barrel plane. The free-energy difference between the open and the closed states of OmpG at pH 5.6 is about 10 kcal/mol. A second metastable conformation appears at $\xi = -3.0$ Å and corresponds to a tightly closed state. In this conformation, loop-1, in addition to loop-6, is oriented towards the β-barrel lumen, and interact by virtue of a salt bridge between E24 of loop-1 and R228 of loop-6. This tightly closed conformation is ~2.5 kcal/mol above the canonical closed state, indicative that an appreciable population of the porin is in chemical equilibrium between the closed and the tightly closed conformations at low pH. While loop-6 is responsible for closing the porin, it has been suggested that other loops can contribute to this conformational transition[4,5]. Our finding of a tightly closed conformation, to the best of our knowledge not documented before, points towards a cooperative role between loops-1 and -6 to block the porin.

Conversely, at neutral pH, only a sharp, single energy minimum is found at $\xi = 1.8$ Å that corresponds to a fully open structure. In sharp contrast with the free-energy landscape at acidic pH, the free energy abruptly increases at lower $\xi$ values, suggesting that in this range of nearly neutral pH, OmpG predominantly exists in an open conformation. These two energy landscapes are in excellent qualitative agreement with

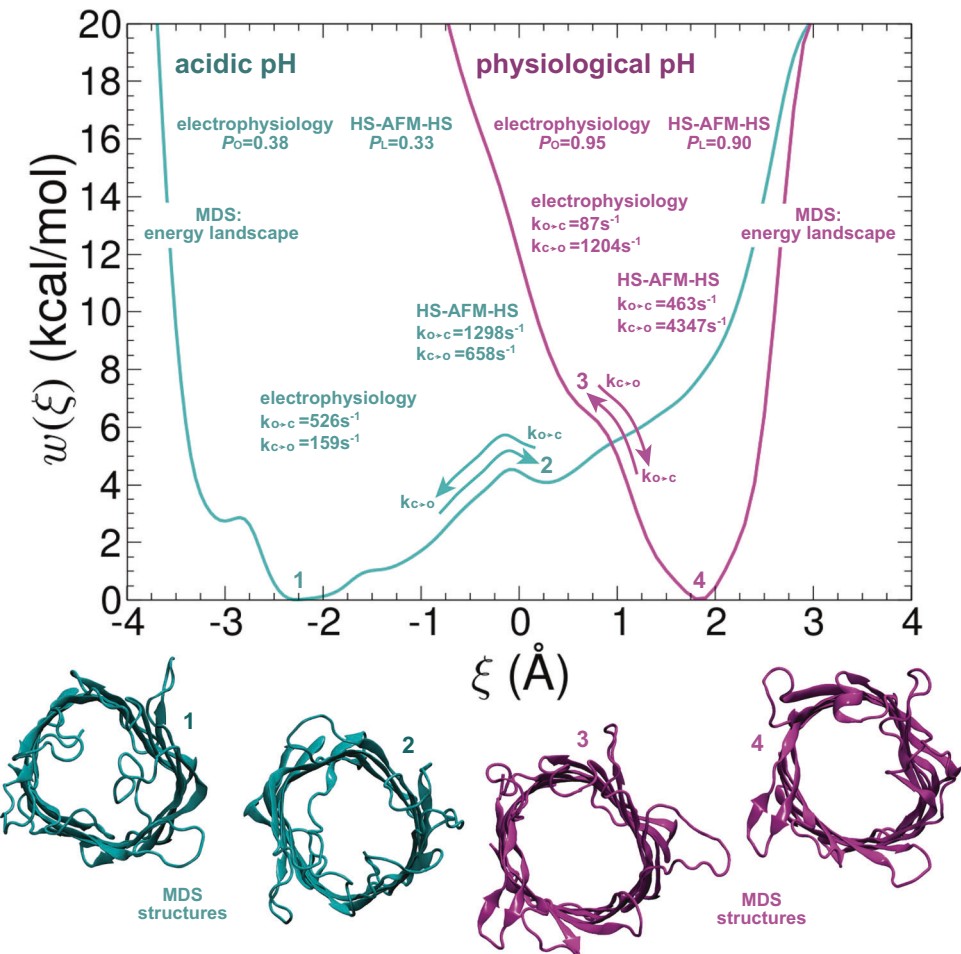

**Fig. 4 OmpG free-energy landscapes.** Free-energy profiles for the gating transition of OmpG at pH 7.5 (magenta curve) and pH 5.6 (cyan curve). Conformations of OmpG for representative basins from the potentials of mean force are depicted: closed (1) and transition state (2) at pH 5.6, open (4) and transition state (3) at pH 7.5. a tightly closed metastable state is found at $\xi = -3.0$ Å where loop-1 interactions stabilize loop-6 closure. Open probabilities and rate constants from both electrophysiology and HS-AFM-HS are indicated (see "Discussion"). Rate constants from electrophysiology refer to the slower gating events that represent loop-6 motions.

electrophysiology at neutral (Fig. 2a–c; Supplementary Fig. 2) and acidic pH (Fig. 2b–d; Supplementary Fig. 3) and HS-AFM-HS (Fig. 2e–h; Supplementary Fig. 4): At neutral pH, the channel is almost always open and displays only occasional flickers to the closed state, while at acidic pH, closed and open states coexist, with a measurable bias in favor of the closed state.

## Discussion

Ion channels undergo conformational changes between open and closed states, i.e. gating. Gating can be modulated by manifold stimuli of biochemical nature such as ligands, pH[33,34], or physical stimuli such as voltage, temperature or force[35–37], and occurs in the millisecond regime[3,38]. Here, we took advantage of the technical advances provided by the HS-AFM-HS[18] and of OmpG as a channel gating model, where conformational dynamics and ion-channel flux gating are related via a simple loop movement, to correlate structural dynamics with function. It was our objective to provide with HS-AFM-HS a tool to measure conformational 'gating' with temporal resolution similar to electrophysiology. From these measurements, we derived steady-state average populations of each state and single-molecule state interconversion kinetics. These experimental efforts are accompanied by MDS that provide an atomic-level understanding of the gating process, and, via enhanced-sampling methods, the underlying free-energy landscapes.

Together, the three methods, HS-AFM-HS, single-channel electrophysiology and MD simulations provide the following picture of the OmpG gating process: At neutral pH, the OmpG pore is almost exclusively open. Electrophysiology ($Po_{(pH7.6)} = 95\%$) and HS-AFM-HS ($P_{L(pH7.6)} = 90\%$) show high propensity for the open state, and MD simulations show that the open state is in a deep and sharp free-energy well. Occasional excursions to the closed state ($k_{O-C(ephys)} = 87 \text{ s}^{-1}$ and $k_{O-C(HS-AFM-HS)} = 463 \text{ s}^{-1}$) are events of only very short duration ($k_{C-O(ephys)} = 1240 \text{ s}^{-1}$ and $k_{C-O(HS-AFM-HS)} = 4347 \text{ s}^{-1}$). At acidic pH, the closed state is favored, as reported by all methods, single-channel electrophysiology ($Po_{(pH5.0)} = 38\%$), HS-AFM-HS ($P_{L(pH5.0)} = 33\%$) and MD simulations ($\Delta G = \sim 8$ kcal/mol), but the open state is often visited ($k_{C-O(ephys)} = 286 \text{ s}^{-1}$ and $k_{C-O(HS-AFM-HS)} = 555 \text{ s}^{-1}$) for $<\tau_{C(ephys)}> = 1.9$ ms and $<\tau_{C(HS-AFM-HS)}> = 0.77$ ms, giving $k_{O-C(ephys)} = 526 \text{ s}^{-1}$ and $k_{O-C(HS-AFM-HS)} = 1298 \text{ s}^{-1}$. Thus, in both pH conditions the structural dynamics detected by HS-AFM-HS indicate faster kinetics than the functional gating-state changes measured by single-channel electrophysiology. We interpret this difference in the following way: HS-AFM-HS captures loop-6 fluctuations in varying positions over the β-barrel where the ion conductive pore is not yet closed to ion passage, thus the measured structural dynamics would be expected to exceed the gating dynamics. Alternatively, it is plausible that OmpG displays slightly different kinetics in the different bilayers, i.e., the

electrophysiology bilayer was constituted of DPhPC and contained decane, which is known to thicken bilayers, whereas the HS-AFM-HS measurements were performed in a POPE:POPG (8:2) bilayer. In this context, it has been previously shown that the presence of charged lipids affected the OmpG gating frequency[39]. For instance, when OmpG was inserted into an asymmetric bilayer from a side that contained negative charges, the gating frequency was increased relative to OmpG in neutral lipid bilayers (DPhPC). These results suggested that interaction of negatively charged lipids with negatively charged extracellular loops increased gating frequency[39]. Since in our HS-AFM experiments negatively charged POPG is present, it is possible that we observe a related effect.

The free-energy differences between the states from electrophysiology and height-spectroscopy are similar at both pH values, though the results from electrophysiology would, based on the slower transition kinetics, suggest higher barriers than height-spectroscopy. We cannot rule out the possibility that the probe may provide thermal energy to the system that may lower the energy barrier between states when analyzed by HS-AFM-HS. At neutral pH, where the closed state is only short-lived, the free-energy difference between the states are $\Delta G_{(ephys)} = 1.7$ kcal/mol and $\Delta G_{(HS\text{-}AFM\text{-}HS)} = 1.3$ kcal/mol. At acidic pH, where both states coexist with similar occurrence, the free-energy difference is smaller, $\Delta G_{(ephys)} = 0.28$ kcal/mol and $\Delta G_{(HS\text{-}AFM\text{-}HS)} = 0.41$ kcal/mol. The free-energy differences from MDS are higher than the experimental values. At neutral pH, the open state is in a very steep energy well, and at acidic pH $\Delta G_{(MDS)} \sim 8$ kcal/mol. We attribute these differences to methodological variances, as well as sampling and force-field inaccuracies. It is, however, notable that the free-energy differences follow the same trend, and that the shape of the free-energy landscapes with a wider and shallower shape at acidic pH, and a steep well at neutral pH match qualitatively well the experimental results. Another interesting finding of the MDS is the tightly closed state, in which, in addition to loop-6, loop-1 is oriented towards the β-barrel lumen, where they interact with each other. Unfortunately, our current HS-AFM and HS-AFM-HS data cannot provide complementary information with regard to loop-1 position and fluctuations, because loop-6 dominates the topography and fluctuations. Future investigations with a loop-1 deletion mutant might resolve kinetic features, conformationally and functionally, that may correspond to this tightly closed state.

HS-AFM-HS allows measuring dynamic biomolecular processes occurring under the tip, from which the diffusion coefficient, surface concentration and oligomeric states of diffusing molecules could be calculated, as well as conformational dynamics[18]. Here, we applied this method to measure conformational signatures of channel gating at up to 100 μs temporal resolution. Single channel electrophysiology is a powerful technique to measure functional dynamics of channel proteins. Such, the two approaches are complementary. Indeed, it is difficult correlate the functional states determined by electrophysiology with structural states. Often, functional studies are correlated with static average structures from X-ray crystallography and cryo-EM. However, while these structures provide crucial high-resolution information about gating residues and selectivity filters, they do not provide intermediate states or kinetic information. The X-ray structures of OmpG at pH 7.5 and pH 5.0 report static open and closed conformations, respectively, while electrophysiology reports that at both pHs both states interchange with different kinetics and overall state probabilities. Here, using HS-AFM-HS, we attempted to bridge this gap and access kinetic and probabilistic information from a conformation angle. HS-AFM-HS has currently several limitations: (i) Instrumental drift makes recordings over long experimental time challenging. Currently, we need to switch between imaging and height

spectroscopy modes to assure recording on a specific location on top of a molecule. (ii) The conformational signal must be larger than the thermal noise of the cantilever, which might become limiting at high bandwidth. (iii) The temporal resolution is limited by the feedback bandwidth, currently at ~100 kHz in ideal conditions. (iv) The sample is substrate supported, in contrast to a free-standing bilayer in single channel recordings. Thus, HS-AFM-HS closes an experimental gap for the analysis of fast biomolecular dynamics but remains improvable with future technological efforts.

Correlating structure, structural dynamics and functional dynamics is a current biophysical challenge. Only when knowing the three, we can assign parameters like turnover efficiencies, or distinguish conformational and functional rates, and build conformational and functional free-energy landscapes. Here, we pave the way towards correlating structural and functional dynamics at the single-molecule level and in a near-native environment.

## Methods

**OmpG purification and reconstitution.** OmpG was expressed in BL21(DE3) pLysE *E. coli* and purified from inclusion bodies. After growing 1-L cell culture to OD of 0.6, expression was induced by adding IPTG to a final concentration of 0.5 mM. Cells were further grown for 4 h and harvested. Cell pellet was dissolved in 10 mL of lysis buffer containing 200 μg/ml hen egg white lysozyme, 10 mM Tris pH 8.0, and 1 mM EDTA and stirred on ice for 30 min. Cell lysis was performed using probe sonication (model CL-18, 80% amplitude, 20 s on/40 s off, 15 min, 4 °C). Lysate was centrifuged at 2,600 x g for 30 mins at 4 °C and pellet was dissolved in 50 mL denaturation buffer (8 M urea, 10 mM Tris-HCl pH 8.0, 0.1 mM EDTA). Urea treated inclusion bodies were spun at 45,000 rpm using type 45 Ti rotor and supernatant was collected and filtered before loading onto a HiTrap DEAE Sepharose column (GE Healthcare). Protein was eluted with 0–150 mM NaCl gradient in denaturation buffer. Purified OmpG was added dropwise into refolding buffer containing 70 mM n-octyl-β-glucopyranoside and refolded at 37 °C for 72 h. Refolded OmpG was reconstituted into POPE-POPG lipids at lipid-to-protein ratios (LPR) 0.5 and 0.7. Briefly, a lipid film was formed in a glass tube by drying POPE:POPG (80:20) lipids in chloroform using argon gas and by leaving overnight in the desiccator. A lipid film was hydrated with 10 mM Tris-HCl pH 7.6, 150 mM Nacl buffer and bath sonicated for 30 min to make lipid vesicles. Then vesicles were solubilized with βOG. Desired quantities of solubilized lipids and protein (for LPRs 0.5 and 0.7) were mixed for several hours at room temperature. Detergent was removed overnight by dialyzing against 10 mM Tris-HCl pH 7.6, 150 mM Nacl.

**HS-AFM imaging.** 1–2 μL of the OmpG reconstituted vesicles were deposited on a 1.5-mm² freshly cleaved mica surface, which was glued with epoxy to the quartz sample stage. After 20–30 min incubation, sample was gently rinsed with imaging buffer (10 mM Tris-HCl pH 7.6, 150 mM NaCl for neutral pH, or 10 mM Tris-HCl or NaOAc pH 5.0, 150 mM NaCl for acidic pH) and mounted in the HS-AFM fluid cell. All images in this study were taken by HS-AFM (Research Institute of Biomolecule Metrology Co.) operated in amplitude modulation mode using optimized scan and feedback parameters. Short (8 μm) cantilevers (USC-F1.2-k0.15, NanoWorld) with nominal spring constant of 0.15 N/m, resonance frequency of 0.6 MHz, and a quality factor of ~2 in buffer were used. HS-AFM movies were collected either at pH 7.6 or pH 5.0 with imaging rate 200 ms.

**HS-AFM data analysis.** HS-AFM movies were drift-corrected using lab-made plugins in imageJ. The average and standard deviation maps shown in Fig. 1b–d and 1g–i were calculated using home-made plugins in imageJ in the following way: (i) A single OmpG dimer molecule is chosen in the data set Fig. 1a and b, respectively. (ii) This single molecule was 2-fold symmetrized and used as reference-1 to search for molecules in the respective HS-AFM data in a first round of cross-correlation searches. (iii) From this first round of cross-correlation searches an average was calculated that serves as reference-2. (iv) This reference was twofold symmetrized and used for a second round of cross-correlation searches. (v) All found particles were extracted, merged in a stack and used to either calculating an average map, where the value of each pixel $x_n y_n$ is the average value of all pixels $x_n y_n$ in all molecules (Fig. 1b–h), or used to calculating a standard deviation map, where the value of each pixel $x_n y_n$ is the standard deviation of all pixels $x_n y_n$ in all molecules (Fig. 1d–j).

**Height spectroscopy and analysis.** High-speed AFM height spectroscopy (HS-AFM-HS) measurements were taken directly after HS-AFM imaging by stopping the x–y piezos, with the z-feedback remaining active. The positioning of the HS-AFM probe when engaging into HS-AFM-HS is at center x-y coordinates (x/2, y/2) of an image with dimensions x, y. Before engaging into HS-AFM-HS, we recorded images at small scan size, typically 30 ×30 nm, centering a single OmpG molecule.

In order to minimize lateral drift, we collected height spectroscopy data for 10 to 30 s, and then switched back to imaging mode, to reposition the tip to the center of mass of an OmpG channel. We note that the HS-AFM tip radius is not as crucial for HS-AFM-HS as it is for imaging. A relatively blunt tip, as long as it can penetrate to some extent into the pore can still report gating dynamics. However, the tip radius ultimately determines the full amplitude that can be measured, depending on how deep it can penetrate into the pore; this should however not influence the measured kinetics.

Measurements were taken with a free oscillation amplitude of ~3 nm and a set-point amplitude of >90% of the free amplitude. Feedback settings were optimized to maximize feedback response speed. The oscillation frequency of the probe is between 500 and 600 kHz in buffer solution. At this oscillation frequency, the oscillation cycle is ~2 μs. However, the time during which the probe is in contact with the sample is only ~10% of the oscillation cycle[16], when the cantilever is in its lower swing, thus only ~200 ns or less. Therefore, the tip interaction is orders of magnitude shorter than the conformational dwell-times measured. The oscillation amplitude is, importantly, larger than the height of the loop-6, and thus the tip retracts during each oscillation far above the space that loop-6 explores. We therefore think that the HS-AFM probe does not prevent loop-6 from evading the barrel lumen. HS-AFM operation is based on the use of ultra-short, 8 μm, cantilevers (USCs, here USC-F1.2-k0.15) with high resonance frequency. Therefore, we cannot test different frequencies, as the HS-AFM operated in amplitude modulation mode excites the cantilever at resonance frequency. However, based on our results where the open and closed events are in the 100 μs to 1 ms time range, oscillation frequency of 2 μs should not be a limiting factor. While we cannot entirely exclude that the tip influences the loop kinetics, we note that charge effects of the tip would be shielded by the electrolyte in the solution (150 mM KCl). Regarding hydrodynamic pressure, the tip (diameter ~2 nm, height of ~2.5 μm) separates the cantilever (oscillating at ~3 nm amplitude) from the surface. The large height of the tip is crucial to eliminate hydrodynamic pressure caused by the swinging oscillating cantilever on the sample[40].

Z-piezo data was captured with home written software and a data acquisition board with a maximum acquisition rate of 2,000,000 samples s$^{-1}$ (LabView programming, NI-USB-6366 card, National Instruments, USA). Height-time traces were analyzed by a state detection algorithm using MATLAB (Matlab, Mathworks, Natick, USA). To determine the states and state transitions, we adapted the Step Transition and State Identification (STaSI) algorithm[29] developed for discrete single-molecule data analysis for our HS-AFM-HS data. To apply the algorithms the HS-AFM-HS data (captured at ~500–600 kHz) was low-pass filtered to 20 kHz and segmented into 0.4–0.6 s time windows to account for baseline drift as baseline fluctuations created problems to the state analysis when longer segments are analyzed. The STaSI algorithm was then used to detect step transitions using Student's t-test before being grouped into segments by hierarchical clustering. An optimum number of two states was suggested by weighing between the complexity of the model and the goodness of the fit ('minimum description length', MDL) to find the simplest model with the least fitting error. Only traces with two states showing a minimum of height difference between assigned states of 2.5 Å were considered in the analysis to prevent incorrect state assignment due to noise (for HS-AFM-HS the height noise is ±1 Å).

**Single channel recordings and analysis.** Single molecule electrophysiology recordings were performed in a horizontal planar lipid-bilayer setup where the upper (cis) and lower (trans) chambers were separated by a partition with a ~100 μm diameter aperture. Buffer solution (1 M KCl and 10 mM Tris-HCl pH 7.6 or pH 5.0 or 1 M KCl and 20 mM sodium acetate pH 5.0) was added to both chambers. 20 μL of 25 mg/ml 1,2-diphytanoyl-sn-glycerol-3phosphocholine (Avanti Polar Lipids) in chloroform was dried with nitrogen gas and dissolved in n-decane to a concentration of ~6.25 mg/ml. Bilayers were painted over the aperture with a smooth capillary glass stick[41]. Refolded OmpG at nanomolar concentration in βOG was added to the cis side of the bilayer. Single channel measurements were performed using an Axopatch 200 A (Molecular Devices) amplifier interfaced to a computer via digitizer Digidata 1440 A (Molecular Devices), and Clampex (Molecular Devices) software. Current traces were recorded at +40 mV with a 5 kHz low-pass filter and 100 kHz sampling rate. Single channel traces (from at least 5 separate bilayers at each pH with recording times in the range of 20–150 s for each channel) were analyzed using Clampfit 10.7 (Molecular Devices) to determine open probabilities and dwell times. Time constants were determined by fitting exponential probability distribution functions to the dwell-time histograms in Clampfit 10.7 (Molecular Devices).

**MD simulations.** All-atom molecular dynamics (MD) simulations were performed with the NAMD program[30] of OmpG in a realistic lipid environment formed by a dimyristoylphosphatidylcholine (DMPC) bilayer in equilibrium with two water lamellae (Supplementary Fig. 5), to characterize the structural features of OmpG in a membrane. The starting point for each simulation was the crystallographic structure of the open and the closed conformations (PDB IDs: 2iwv and 2iww, respectively), and the protonation states of the titratable residues (Supplementary Table 3) were assigned according to the pH used to crystallize each conformation, employing the program propka 3.1[42]. Cl$^-$ and K$^+$ counter ions were added to ensure electric neutrality and a final salt concentration of 150 mM. Periodic

boundary conditions were applied. Molecular interactions were described with the all-atom CHARMM36 force field for biomolecules[43,44]. The van der Waals and short-range electrostatic interactions were computed within a 12 Å cut-off, while long-range electrostatic interactions were evaluated using the particle mesh Ewald (PME) algorithm[45]. A time step of 4 fs was used to integrate the equations of motion, with a hydrogen-mass repartitioning scheme, and the RATTLE algorithm[46] was employed to constrain the equilibrium length of all covalent bonds containing hydrogen atoms. The temperature and pressure were maintained at 298 K and 1 atm, respectively, using Langevin dynamics and the Langevin piston algorithm[47]. The following equilibration protocol was utilized, prior to collecting 3 μs of unrestrained sampling for analysis purposes: (i) 1,000 steps of energy minimization with frozen heavy atoms of the protein and crystallographic-water oxygen atoms, (ii) 50 ns of sampling with harmonic restraints (force constant of 1 kcal/mol Å$^2$) on all heavy atoms of the protein (free crystallographic water molecules), (iii) 50 ns of sampling with harmonic restraints (force constant of 1 kcal/mol Å$^2$) on the backbone atoms of the protein, and (iv) 20 ns of sampling, whereby the harmonic restraints on the backbone atoms were smoothly removed. Backbone atoms gained a 10% flexibility every 2 ns of sampling. Both conformations of OmpG are remarkably stable throughout the MDSs. This is evidenced by low fluctuations (< 1.5 Å) in the time series of the root mean square deviation (RMSD) over backbone atoms for the 14 β-strands (Supplementary Fig. 6). The barrel-shape of the porin is fully preserved in both simulations, and there is no noticeable loss of tertiary structure. Furthermore, the root-mean square fluctuations (RMSF) over the C$_\alpha$ atoms were determined as a second metric of structural stability. These computed quantities can be readily compared with experimental RMSFs inferred from the crystallographic B-factors for both the open and the closed conformations (Supplementary Fig. 7). The theoretical and experimental RMSF profiles agree qualitatively, with low values corresponding to residues pertaining to the β-strands, and higher values to residues in turns and loops. It is also noteworthy that the secondary structure in both conformations is fully preserved during the entire MD simulations (Supplementary Fig. 8). In particular, there is no loss of β-strand in the porin. All rendering and MD analyses were performed with the VMD software[48].

**Reporting summary.** Further information on research design is available in the Nature Research Reporting Summary linked to this article.

## Data availability

Data supporting the findings of this manuscript and the MATLAB codes used for analysis are available from the corresponding author upon reasonable request. Source data are provided with this paper.

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

## Acknowledgements

The work presented here was supported by the NIH NINDS (R01NS110790 to S.S.), American Heart Association (18POST34080488 to R.R.S.G), the Agence Nationale de la Recherche (grant ProteaseInAction to C.C.), and the European Regional Development Fund (ERDF to C.C.).

## Author contributions

R.R.S.G and S.S designed research. R.R.S.G expressed, purified, and reconstituted protein. R.R.S.G and Y.J performed HS-AFM experiments. R.R.S.G and G.R.H analyzed HS-AFM data. R.R.S.G, X.G and C.N performed and analyzed single channel recordings. S.S supervised HS-AFM, single channel recording experiments, and data analysis. J.J.M and C.C performed and analyzed MD simulations. R.R.S.G, J.J.M, C.C, C.N and S.S wrote the manuscript.

## Competing interests

The authors declare no competing interests.
