## [Peer Review File · Nature Communications]

Reviewers' Comments:

Reviewer #1:

Remarks to the Author:

In the work by Gari R. et al. showed the use of high speed-atomic force microscopy-height spectroscopy (HS-AFM-HS) to study the structural dynamics of an outer membrane protein G (OmpG) channel. In HS-AFM-HS, the AFM tip placed at fixed x-y coordinates oscillates at the z-axis with an amplitude of 3 nm. This approach allows the data (height at the z-axis) to be acquired at a maximal sampling rate of 0.5 us, compared with the AFM imaging rate of 100 ms. OmpG is a membrane channel that shows a pH-dependent gating, the mechanism of which was well studied by the single-channel electrophysiological approach. By fixing the AFM tip's x-y position, the HS-AFM-HS measures the height variation induced by the structural changes of the flexible loops of a single OmpG channel. While the HS-AFM-HS approach could potentially be used for directly observing the structural dynamics of a membrane protein channel at the high-temporal resolution, there is a lack of discussion about what additional information about the OmpG gating has been revealed by HS-AFM-HF that was not previously learned by single-channel recording. In addition, because the HS-AFM-HS is essentially a single-molecule approach, the experiments should include more rigorous statistical analysis about the single-channel variation. The detailed comments are listed below:

1. Figure 1. the interpretation of the HS-AFM imaging of OmpG is debatable. It seems to this reviewer another possibility to interpret the topography is to fit the highest region with loop 3 and 4 as the recent NMR structure of OmpG in lipid bilayer showed an extended barrel structure at loop 3 and loop 4 while other regions had a shorter barrel (Retel JS et al. 2017, <https://doi.org/10.1038/s41467-017-02228-2>).
2. HS-AFM-height spectroscopy recordings: it is not clear how the x-y coordination of the probe was decided. Was the probe placed at the center of the OmpG channel? What is the resolution of the probe at the x-y surface? How does the position of the probe affect the recording? For example, could the height spectroscopy trace vary with different locations of the tip: a tip positioned at the center of the channel versus one closer L6?
3. What is the oscillation frequency of the probe? Would the tip affect the dynamics of a flexible polymer? ? One can imagine when the tip is inside of the barrel, it will prevent L6 from evading into the lumen; therefore the tip could preferentially select an open state. How does the oscillation frequency of the tip affect the height spectroscopy trace of the OmpG pore?
4. How long can a single membrane protein channel be recorded for? Does the tapping tip have an impact on the structural stability of the membrane protein channel?
5. Figure S4, the authors showed two traces of height spectroscopy recordings at pH 7.5 at different time scales (4s segments vs. 0.4s segments). Are these two traces from the same recording of an OmpG channel or two separated recordings? If the two traces were from the same recording, can the author explain why the histograms of the two traces vary so much? By estimation, the open probability of the Fig S4b trace could be around or smaller than 85%, much less than the 95% of the Figure S4a. Similarly, the two traces at pH 5.0 have shown an even greater difference than those at pH 7.5. The open probability of the Fig. S4c trace could be approximately 10% by estimation.
6. Figure 2c: the dwell time histograms of the closed states at pH 7.6 and 5.0 showed two well-separated peaks indicating two types of gating events at different time scales. However, only a single time constant was shown for both cases. Please include the error for the time constant calculation. Also, please clarify which population of the event was used to derive the time constant.
7. Figure 2: HS-AFM height spectroscopy recordings of OmpG. At pH 7.6, the height of the open state was around 0.5 nm and the closed state was above 1.0 nm. In contrast, at pH 5.0, the height of the open state was closed to 0.2 nm and the open state was at the 0.5 nm level. Why is there such a large difference in the height of the open and closed states at the two different pH conditions? How was the height of the tip pre-calibrated? Again, in Figure S4, the height levels of the open and closed states at pH 7.5 and pH 5.0 are also different. Moreover, the relative height levels of the open and closed states at pH 7.5 and pH 5.0 are also inconsistent. At pH 7.0, the height changes by about 1.0 nm from the open to the closed state. At pH 5.0, the height changed by 0.5 nm for one channel while the other channel showed a 0.3 nm shift in height. This raises a concern about the criteria to assign

the height levels of open and closed states for a recorded trace without referencing to prior knowledge (or when no prior knowledge is available). If the height level 1.0 nm of Figure S4a trace was assigned as the open state, why the same height level of the Figure S4c trace was assigned as the closed state?

8. Line 160: the time constants of the open and closed states have no errors. How many OmpG channels were recorded, and what are the variances from channel to channel?

9. Page 11 line 302. Different lipid conditions may alter the kinetics. The effect of charged lipids on the OmpG gating was studied by the work below, please cite and discuss the result. (William L. Hwang, Min Chen, Bríd Cronin, Matthew A. Holden, and Hagan Bayley, Asymmetric Droplet Interface Bilayers, *Journal of the American Chemical Society* 2008 130 (18), 5878-5879, DOI: 10.1021/ja802089s)

10. If the primary goal of the work is to demonstrate the use of the HS-AFM-HS technique for studying the structural dynamics of a membrane channel, please include detailed discussions about the advancement and the limitation of the HS-AFM-HS, particularly as a comparison to the single-channel electrophysiological approach.

Minor:

1. Page 11 line 329. *E. coli* should be italic
2. Line 343: "Nacl" should be "NaCl"
3. Figure S4: The pH of a and b is shown as 7.5 while Figure 2 is pH 7.6. Which one is correct?
4. Table S3: "Protonation states for critical residues are highlighted in red." No residue was highlighted.

Reviewer #2:

Remarks to the Author:

This paper by Gari et al. describes the use of combined high speed Atomic Force HS-AFM-HS, single channel electrophysiology and molecular dynamics simulations to monitor conformational dynamics in OmpG. Gari et al. correlate structural changes observed using HS-AFM-HS to channel dynamics from electrophysiology measurements.

HS-AFM imaging of OmpG in lipid bilayers at 10 frames per second:

This section describes the structural changes in OmpG observed using HS-AFM. These measurements correlate with those in the literature. There main points as I read this section are; that the substructure of OmpG is visible by HS-AFM including the protrusions of loop 2 and 6 are visible by HS-AFM, that the flexibility of loop 6 – the key mobile & functional structure of this pore can be mapped by HS-AFM, and that there appears to be a visible difference in the dimeric OmpG structures between the open and closed states. I have a few questions to clarify the conclusions being drawn from these data, and would

I believe a number of modifications to Figure 1 would make direct comparisons were drawn between the two structures simpler to see.

How does the outline/correlation average for the closed state compare with that of the open state?

Why is the height so much lower in the closed state? Is this an artefact of the low pH?

Could clearer comparisons between standard deviation measurements of loops and correlation averaged structures be shown? This could include rotating these to be at the same angle to facilitate easier comparison. Interesting observations include the potential stabilization of loop 6 in the centre of the dimer for the closed state, and potential increase of mobility in other loops e.g 2 in the closed state?

Figure 1 minor observations:

Arrowhead in t=11.8s: loop-6 fluctuating over the lumen. Arrowhead in t=12.0s: Fully open state.

I can't see the arrowheads described.

c) Correlation average of OmpG dimers.

How was this outline generated?

h) Correlation average of OmpG dimers. For comparison, the topography outline of the open state (c) is shown (the position of loop-6 is indicated by the asterisk)

I find this confusing – the overlay in c is of the same data set, and here it is from the dataset in c but rotated? This needs to be made clearer or changed to show the topography outline of the open state.

i) Standard deviation (std) map (n=2472) from the averaging process in (c)

Is this (g)?

Loop-2 (L2) does not form a β -strand in the closed state, in agreement

In agreement with what?

Correlation between ion flux and loop-6 dynamics of OmpG in lipid bilayers

The single-channel recordings carried out showed good agreement with the literature. Gari et al. observe dwell-times in the millisecond time range, and use HS-AFM-HS to increase the time resolution of their AFM setup to capture these transitions. Gari et al. use the tip position as an indicator of the pore state (open vs closed), with a height change of 5Å observed between the two. Gari et al. report a difference in kinetics as observed by HS-AFM-HS, vs single-channel recordings. The open state time constant is 5x shorter in HS-AFM-HS measurements, and the rest are 2-4 times shorter. The authors attribute this to that loop-6 structural transitions occurring faster than the corresponding functional state transitions in electrophysiology. However the authors do not mention the impact of the tip here. Could the tip impact this transition? How do the authors address the presence of the tip, both in terms of local charge, flow and fluid dynamics?

In Figure 2, the closed states appear bimodal, and in the HS-AFM-HS, these second peaks are no longer visible, could this be due to the impact of the tip, or due to the increase in noise of the HS-AFM-HS measurements?

Overall as the next section shows, some of the HS-AFM-HS measurements could show 'closed' states for those which are partially open, and I would encourage them to clarify/moderate their structural observations by HS-AFM-HS (e.g, the use of apparent in the next section and compare these to other measurements. e.g. NMR data to gauge the likelihood of these AFM data revealing the 'true' conformation.

Analysis of OmpG gating from atomistic simulations

Gari et al. then use MD simulations to gain atomistic structural insights into the dynamics and interactions of loops in OmpG. This section provides structural information to explain why the HS-AFM-HS disagrees with the electrophysiology measurements. This section expands on the flexibility of loop 6 observed in Figure 1 using HS-AFM, to provide atomistic detail of partially closed structures, explaining the differences in structural/functional studies using electrophysiology and HS-AFM-HS in Figure 2. Can the authors reflect on this, linking flexibility to conductance and linking observations in Fig 1d/I to the MD simulations?

Some of the figure references in this paragraph are hard to follow – could the authors check through (e.g. compare Figure 1e with Figure 1a and single reference to Figure 1e) If these are correct could the authors provide more information as to what is being compared/shown?

Free-energy landscape of OmpG gating

Gari et al. then bring together their AFM measurements and MD simulations to define the free energy landscape of OmpG gating, with the main finding being that a tightly closed conformation, formed through a cooperative partnership between loops-1 and -6 works to block the porin. The authors do not reference the position of loop 1 in their HS-AFM measurements. I would be interested to know if they can observe, or calculate the position of loop one in their HS-AFM measurements, and see any local stabilization of loop 6 via images, or standard deviation maps.

The discussion of the paper is balanced, and reflects on the use of the three techniques to access currently hard to carry out complete biophysical characterisation of molecular structures. The statement regarding discrepancies between HS-AFM-HS and electrophysiology below could be modified to address the presence of the tip in HS-AFM-HS and its potential to lower the barrier.

"The free-energy differences between the states from electrophysiology and height-spectroscopy are similar at both pH, though the results from electrophysiology would, based on the slower transition kinetics, suggest higher barriers than height-spectroscopy."

Finally – as outlined above – I would be interested to see if the authors see any evidence of this tightly closed state below through HS-AFM imaging or standard deviation measurements as discussed in Figure 1.

"Another interesting finding of the MDS is the tightly closed state, in which, in addition to loop-6, loop-1 is oriented towards the β -barrel lumen, where they interact with each other. Future investigations might resolve kinetic features, conformationally and functionally, that may correspond to this tightly closed state."

Reviewer #3:

Remarks to the Author:

The authors of this well written paper employed high-speed atomic force microscopy height spectroscopy (HS-AFM-HS) and electrophysiology to characterize the open-closed gating of the outer membrane protein G (OmpG) porin of *E. coli*. The experimental results are complemented with atomistic molecular dynamics simulations. The overall goal was to explain the previously well-established pH-dependent gating of this porin by structural measurements of the loop dynamics of OmpG, particularly loop 6, which is known to play a major role in opening and closing the pore of this beta-barrel protein.

HS-AFM-HS is an excellent technique to follow the motions of loop 6 in the gating process with adequate time and spatial resolution. The presented experiments are of high technical quality and demonstrate in this well-chosen model system that the technique is suitable to interpret the electrophysiological data by monitoring height fluctuation on an Angstrom length scale with 10-50 microsecond time resolution. The height fluctuations adequately interpreted as loop 6 motions nicely correlate with the electrophysiological opening and closing dynamics at neutral and acidic pH showing motions in the 100 microsecond to 2 millisecond range. Although the MD simulations qualitatively support the experimental results, the technical limitation to 3 microsecond total simulation times limit a direct quantitative comparison with the experimental data on the same time scale.

The authors are encouraged to consider the following points in a revision:

1. Loop 2 is seen in the averaged AFM contours in Fig 1c (neutral pH), but not in Fig 1h (acidic pH). Perhaps this is real and the crystal structure contour in Fig 1j is due to crystallization conditions. Commenting on the choice of these contours will be helpful.
2. Zhuang et al. (JACS 135, 1501) conducted a comprehensive study of the structural dynamics of OmpG by NMR, related it to electrophysiological data like in the current work, and found by paramagnetic relaxation enhancement measurements and ensemble averaging that loop 6 and to a lesser degree other loops (e.g. loop 2) moved between different conformations in a correlated fashion. It would be interesting to compare this work with the current work in the Discussion and also mention this in the Introduction when discussing that x-ray structures were able to distinguish between open and closed forms of OmpG.
3. The MD simulations were run for a total of 3 microseconds and the simulation analysis of OmpG gating is based on residue R228 (loop 6) motions in this time window. However, NMR relaxation dispersion results of loop 6 residues in Zhuang et al (Supp Fig 1) show that these residues move on a ~500 microsecond timescale (reassuringly very close to the time scale observed by HS-AFM-HS!). Therefore, loop 6 may undergo larger excursions than sampled in the current work by MD. This should be acknowledged. Larger and slower motions of this (and other smaller) loop(s) would still be

consistent with the observed HS-AFM-HS data, but their interpretation should not be constrained to the 3 microsecond window that was accessed by the current MD simulations. – Even so, the simulations are certainly valid and help with the interpretation of the paper, but perhaps in a more qualitative than in a detailed quantitative fashion.

Minor:

4. Lipid:protein ratios are said to be 0.5-0.7 in the AFM samples. It may be more informative to give those as molar rather than weight ratios.
5. Under these L:P conditions, are most proteins observed in crystalline arrays by AFM and do most observed sample areas show crystals? Are there also many areas with non-crystalline proteins or bare membrane areas?
6. The nomenclature of PsubL and PsubO on p. 5 is not clear.
7. In Fig 2a,b,e,f, please explain the red idealized traces that track the gray raw data.
8. Fig 2e shows height transitions at pH 7.6 of about 5Å, but they are smaller (~2Å) in Fig 2f at pH 5, contrary to the statement on p5, l.147.
9. Fig 4: “physiological” should be called “neutral” pH. It is not clear what physiological means for the outer environment of E. coli.
10. Methods, height spectroscopy: if I understand correctly, the cantilever xy scan is stopped at some point after imaging over a crystalline array before switching to height mode and this stop would have to occur over the central hole of the barrel for the method to work. So, what is the success rate of stopping in a successful place over a single OmpG protein vs any other random place? – Or, is this always successful as long as you are over a crystalline array?

REVIEWER COMMENTS

Reviewer #1 (Remarks to the Author):

In the work by Gari R. et al. showed the use of high speed-atomic force microscopy-height spectroscopy (HS-AFM-HS) to study the structural dynamics of an outer membrane protein G (OmpG) channel. In HS-AFM-HS, the AFM tip placed at fixed x-y coordinates oscillates at the z-axis with an amplitude of 3 nm. This approach allows the data (height at the z-axis) to be acquired at a maximal sampling rate of 0.5 us, compared with the AFM imaging rate of 100 ms. OmpG is a membrane channel that shows a pH-dependent gating, the mechanism of which was well studied by the single-channel electrophysiological approach. By fixing the AFM tip's x-y position, the HS-AFM-HS measures the height variation induced by the structural changes of the flexible loops of a single OmpG channel.

While the HS-AFM-HS approach could potentially be used for directly observing the structural dynamics of a membrane protein channel at the high-temporal resolution, there is a lack of discussion about what additional information about the OmpG gating has been revealed by HS-AFM-HF that was not previously learned by single-channel recording. In addition, because the HS-AFM-HS is essentially a single-molecule approach, the experiments should include more rigorous statistical analysis about the single-channel variation. The detailed comments are listed below:

We thank the reviewer for the thorough assessment of our work. As the Reviewer points it out, we used OmpG as a model system to explore the possibilities of HS-AFM-HS for the analysis of membrane protein structural dynamics.

We regret that we have not been able to emphasize sufficiently what additional or different (complementary) information can be gained from HS-AFM-HS as compared to electrophysiology. In a nutshell, the electrophysiology measurements report about ion flow through the channel—or not, i.e. gating. However, the information about the ion flow does not necessarily relate directly to the conformational dynamics. It was our objective to provide with HS-AFM-HS a tool to measure conformational ‘gating’ at similar temporal resolution as electrophysiology (lines 321-322).

Our results show that OmpG undergoes faster conformational fluctuations as compared to single channel recordings. MD simulations on the microsecond timescale echo the HS-AFM-HS results. HS-AFM-HS captures loop-6 fluctuations in varying positions over the β -barrel, in a series of configurations where the ion conductive pore is not necessarily closed to ion passage. Thus, the measured structural dynamics exceed the functional gating dynamics.

Regarding the single-molecule analysis: We have analyzed HS-AFM-HS data carefully using the STaSI algorithm, which provides unbiased step-transition detection and hierarchical segment clustering. As a result, the algorithm proposes an optimum number of states, weighing between the complexity of the model and the goodness of the fit (‘minimum description length’, MDL). We further enriched our supplementary table 2, in which we now report about the individuality of the OmpG molecules analyzed.

1. Figure 1. the interpretation of the HS-AFM imaging of OmpG is debatable. It seems to this reviewer another possibility to interpret the topography is to fit the highest region with loop 3 and 4 as the recent NMR structure of OmpG in lipid bilayer showed an extended barrel structure at loop 3 and loop 4 while other regions had a shorter barrel (Retel JS et al. 2017, <https://doi.org/10.1038/s41467-017-02228-2>).

The Referee brings up a valid point regarding our assignment of the AFM topography (Figure 1) with respect to the molecular structure. As the Referee can see in our figure 1b, in AFM, the OmpG barrel (open state) has one side that displays high protrusion, and one side that displays low protrusions. A close look at the X-ray structure (open state) reveals low protruding features on one side Loop-7, Loop-1, Loop-2 (see sideview). Thus, this global assessment basically constrains the barrel placement to one position. In

addition, this placement sets Loop-2 precisely at the location where AFM detects a laterally protruding 'nose'. Loop-2 forms a continuation of the β -strands in the X-ray structure, which is a sturdy secondary structure element that we would expect to give a good signal in AFM. Finally, this placement, places Loop-6 to the highest protrusion, which in addition is the region that showed largest mobility in the SD-map. Thus, based on these four evidences, there was literally only one reasonable way to place the molecular structure into our topography.

Left: Sideview on OmpG: Loop7, facing right, Loop-1 and Loop-2 facing the viewer are protruding much less than loops on the other loops.

Right: Topview on OmpG: Loop-2 forms two β -strands that laterally protrudes from the barrel circumference.

These features, together with Loop-6 being the most protruding, served to assign the HS-AFM average topography to the molecular structure, giving only one reasonable placement.

Our assignment of the closed structure is almost entirely based on the lattice lines as defined by the open structure.

We think, however, that the Referee makes a justified point, regarding the NMR structures in Retel et al. We note that Loop-4 is still in the high protrusion area of our topography. Maybe, Loop-3 is too floppy to give a strong topographical signal in AFM imaging.

In response, we have added a clarifying paragraph that our assignment is just the most likely structural correspondence. We also refer to the NMR structure paper and discuss the location of Loop-3 and Loop-4. We note that the precise structural alignment is not of significance for the remainder of the manuscript (lines 104-115)

2. HS-AFM-height spectroscopy recordings: it is not clear how the x-y coordination of the probe was decided. Was the probe placed at the center of the OmpG channel? What is the resolution of the probe at the x-y surface? How does the position of the probe affect the recording? For example, could the height spectroscopy trace vary with different locations of the tip: a tip positioned at the center of the channel versus one closer L6?

The positioning of the HS-AFM probe when engaging into HS-AFM-HS is at the center x-y coordinates ($x/2$, $y/2$) of an image with dimensions x , y . Before engaging into HS-AFM-HS, we recorded images at small scan size, 30×30 nm, thus centering a single OmpG molecule where the HS-AFM probe is placed.

It is difficult to define the resolution of the probe. However, as the Referee can see based on the images (Figure 1), the probe can resolve details on the structure of the OmpG channel. Conceptually, the probe resolution is not such an important topic for HS-AFM-HS. One can imagine that a blunt hemisphere will still report height changes, as long as the topic of interest is the most protruding structural element (which we believe is the case here), and the tip can still penetrate to some extent into the pore. Thus, the tip radius will ultimately influence the power of the amplitudes recorded, but should not influence the kinetics, which is our main objective here. Based on our imaging, we tested that all probes used for HS-AFM-HS could

resolve OmpG as a pore-like molecule, and, therefore, give the strongest topographical signal between an open and a plugged pore.

Of course, there is lateral stage drift. Thus, as a function of time, the probe will experience many places on the OmpG 2D-crystal, even those of no interest, e.g., between molecules. At these locations, there are essentially no height changes detected, until the lateral drift brings the tip back on top of a channel. Since the height difference is most important for Loop-6 plugging the central pore—or not ($>5\text{\AA}$) we only analyzed periods in our HS-AFM-HS height-time traces that displayed conformational fluctuations larger than $\sim 2.5\text{\AA}$ to eliminate nonspecific height changes. In order to minimize drift issues, we typically collect height spectroscopy data for 10 to 30 seconds and go back to imaging mode, to replace the tip to the center of mass of an OmpG channel.

We have added in the materials and methods section some methodological details regarding the placement of the HS-AFM probe, and associated experimental details with respect to the cantilever and tip properties (lines 431-457).

3. What is the oscillation frequency of the probe? Would the tip affect the dynamics of a flexible polymer? One can imagine when the tip is inside of the barrel, it will prevent L6 from evading into the lumen; therefore the tip could preferentially select an open state. How does the oscillation frequency of the tip affect the height spectroscopy trace of the OmpG pore?

The oscillation frequency of the probe varies between 500 and 600 kHz in buffer solution. At this oscillation frequency, the oscillation cycle is $\sim 2\ \mu\text{s}$. However, the time during which the probe is in contact with the sample is only $\sim 10\%$ of the oscillation cycle, when the cantilever is in its lower swing, thus only $\sim 200\ \text{ns}$ or less. Importantly, the oscillation amplitude is larger than the height of Loop-6, and, thus, during every oscillation cycle, the tip moves far above the space that the loop explores. We, therefore, think that the HS-AFM probe does not prevent L6 from evading the lumen.

HS-AFM only works with Ultra-Short Cantilevers (USC), $8\ \mu\text{m}$ short and high resonance frequency. We cannot test different frequencies, as we have to operate the cantilever at resonance. We reason also—based on the results where the open and closed events are in the $100\ \mu\text{s}$ to $1\ \text{ms}$ time range—that the probing with oscillation frequency of $2\ \mu\text{s}$ should not be a limiting factor.

We added a paragraph detailing the time-scale ($\sim 200\ \text{ns}$) of the probe-OmpG interaction with respect to the characteristic timescales of the channel's conformations. We also comment on the probe oscillation amplitude, because it is important to detail that the probe retracts beyond the height of Loop-6 in every oscillation cycle (lines 442-457).

4. How long can a single membrane protein channel be recorded for? Does the tapping tip have an impact on the structural stability of the membrane protein channel?

In principle, we can image or collect height spectroscopy for hours on a membrane without damaging the sample. However, the time-limiting aspect in HS-AFM-HS is the lateral drift, meaning for how long will the tip stay on the same $\sim 2 \times 2\text{-nm}$ region, the area of a channel pore. We are confident that we can probe the exact same pore for several seconds (see Supplementary Figures), after which, we re-center a channel in imaging mode and switch X and Y scanning off again to record HS-AFM-HS traces (lines 431-439).

5. Figure S4, the authors showed two traces of height spectroscopy recordings at pH 7.5 at different time scales (4s segments vs. 0.4s segments). Are these two traces from the same recording of an OmpG channel or two separated recordings? If the two traces were from the same recording, can the author explain why the histograms of the two traces vary so much? By estimation, the open probability of the Fig S4b trace could be around or smaller than 85%, much less than the 95% of the Figure S4a. Similarly, the two traces at pH 5.0 have shown an even greater difference than those at pH 7.5. The open probability of the Fig. S4c trace could be approximately 10% by estimation.

The Referee is right—the 4s and 0.4s segments of traces are from different OmpG channels at both pHs. We revised supplementary table 2 to further highlight the individuality of the single-molecule experiments. Indeed, two of them had open probabilities of 0.85 and 0.89 at pH 7.6. Open probabilities can vary between channels (and likely with time). Regarding—pH 5.0 in Fig.S4c and Fig.S4d, the trace presented in Fig. S4c was divided into segments of ~0.4s, because of a baseline drift. All open probabilities were calculated from the state assignments by the STaSI algorithm on height spectroscopy traces. However, the algorithm struggled assigning states in extended traces when a baseline drift occurred, because the amplitude of the conformational change is only on the order of 5Å, and the baseline drift can be of such a magnitude. The trace in Fig.S4c contains a baseline drift of over 4s, and the overall height histogram, therefore, convolves the smaller amplitude excursions to the open state, as are better visible when histograms from short segments are visualized as in Fig.S4d.

In our revision of the manuscript, we provide in Supplementary Table 2 additional detail in the individuality of the channels, and describe the need to divide long traces into segments, because fluctuations in the baseline created problems to the analysis software (Lines 463-465).

6. Figure 2c: the dwell time histograms of the closed states at pH 7.6 and 5.0 showed two well-separated peaks indicating two types of gating events at different time scales. However, only a single time constant was shown for both cases. Please include the error for the time constant calculation. Also, please clarify which population of the event was used to derive the time constant.

The Referee is right—there are two types of gating events in the dwell-time distribution in 2c (also in 2d). As we mentioned it in the text, we associate the longer dwells (slower gating events) as conformationally relevant events with respect to fluctuations of the entire Loop-6, while we name the shorter events (100µs and shorter than our bandwidth) ‘flickers’ (i.e., short interruptions of the ionic current possibly due to side-chain fluctuations in the barrel). In our analysis, we report about the longer gating component.

In our revision, we provide errors for each time constant (lines 148-151), and state in caption of figure 4 that the rate constants refer to the slower gating events with respect to Loop-6 motions for both techniques, and not to the short flickers detected in electrophysiology (commented in lines 154-156).

7. Figure 2: HS-AFM height spectroscopy recordings of OmpG. At pH 7.6, the height of the open state was around 0.5 nm and the closed state was above 1.0 nm. In contrast, at pH 5.0, the height of the open state was closed to 0.2 nm and the open state was at the 0.5 nm level. Why is there such a large difference in the height of the open and closed states at the two different pH conditions? How was the height of the tip pre-calibrated? Again, in Figure S4, the height levels of the open and closed states at pH 7.5 and pH 5.0 are also different. Moreover, the relative height levels of the open and closed states at pH 7.5 and pH 5.0 are also inconsistent. At pH 7.0, the height changes by about 1.0 nm from the open to the closed state. At pH 5.0, the height changed by 0.5 nm for one channel while the other channel showed a 0.3 nm shift in height. This raises a concern about the criteria to assign the height levels of open and closed states for a recorded trace without referencing to prior knowledge (or when no prior knowledge is available). If the height level 1.0 nm of Figure S4a trace was assigned as the open state, why the same height level of the Figure S4c trace was assigned as the closed state?

The Referee makes a good point. These height values are not absolute, but relative. In reality, as shown in Supplementary Figure 1, the mica level should be at 0 nm, the surface of the membrane at ~3.7nm, and the uppermost surface of OmpG at ~4.8nm. Thus, the Y-axis values in the HS-AFM-HS traces are merely relative height values with respect to the structural element where the tip landed upon engaging HS-AFM-HS mode. Beside the placement, the height difference between states also strongly depend on the tip sharpness, as the sharper tip can probe deeper inside the tight β -barrel, while the blunter tip cannot.

In our revision, we provide additional experimental details regarding the height values in HS-AFM-HS, and indicate that the tip radius ultimately limits the full amplitude that can be measured in these experiments (lines 436-439).

8. Line 160: the time constants of the open and closed states have no errors. How many OmpG channels were recorded, and what are the variances from channel to channel?

In our revision, we have added errors to the time constants (lines 188-190). In HS-AFM-HS, we have recorded five channels at pH 7.6 and three channels at pH 5.0, and as we mentioned above, we had to segment the individual channel traces on account of a baseline drift (lines 463-465). Further information is now provided in Supplementary Table 2.

9. Page 11 line 302. Different lipid conditions may alter the kinetics. The effect of charged lipids on the OmpG gating was studied by the work below, please cite and discuss the result. (William L. Hwang, Min Chen, Bríd Cronin, Matthew A. Holden, and Hagan Bayley, Asymmetric Droplet Interface Bilayers, Journal of the American Chemical Society 2008 130 (18), 5878-5879, DOI: 10.1021/ja802089s)

We thank the Reviewer for bringing to our attention this question and a paper that shows the effect of ~~the~~ charged lipids on the gating behavior of OmpG. We incorporated this information in our discussion and cited the paper. One would hypothesize that charged lipids interact with the charges borne by Loop-6 (lines 342-348).

10. If the primary goal of the work is to demonstrate the use of the HS-AFM-HS technique for studying the structural dynamics of a membrane channel, please include detailed discussions about the advancement and the limitation of the HS-AFM-HS, particularly as a comparison to the single-channel electrophysiological approach.

We thank the Referee for this suggestion, and we have added the following in the discussion section:

HS-AFM-HS allows the dynamic biomolecular processes occurring under the tip to be measured, from which the diffusion coefficient, surface concentration and oligomeric states of diffusing molecules can be determined, as well as the conformational dynamics¹⁸. Here, we applied this method to measure conformational signatures of channel gating at up to 100 μ s temporal resolution. Single-channel electrophysiology is a powerful technique to measure functional dynamics of channel proteins. As such, the two approaches are complementary. Indeed, it is difficult correlate the functional states determined by electrophysiology with structural states. Functional studies are often correlated with static average structures from X-ray crystallography and cryo-EM. However, while these structures provide crucial high-resolution information about gating residues and selectivity filters, they do not provide intermediate states or kinetic information. Taking OmpG, for example, the X-ray structures at pH 7.5 and pH 5.0 report static open and closed conformations, respectively, while electrophysiology reports that, at both pHs, both states interchange with different kinetics and overall state probabilities. Here, using HS-AFM-HS, we attempted to bridge this gap and access kinetic and probabilistic information from a conformation angle. HS-AFM-HS has currently several limitations: (i) Instrumental drift makes recordings over long experimental time challenging. Currently, we need to switch between imaging and height spectroscopy modes to assure recording on a specific location on top of a molecule. (ii) The conformational signal must be larger than the thermal noise of the cantilever, which might become limiting at high bandwidth. (iii) The temporal resolution is limited by the feedback bandwidth, currently at ~100kHz in ideal conditions. (iv) The sample

is substrate supported, in contrast to a free-standing bilayer in single channel recordings. Thus, HS-AFM-HS closes an experimental gap for the analysis of fast biomolecular dynamics, but remains improvable with future technological efforts. (lines 367-387; last paragraph in the discussion section).

Minor:

1. Page 11 line 329. E. coli should be italic

Fixed

2. Line 343: “Nacl” should be “NaCl”

Fixed

3. Figure S4: The pH of a and b is shown as 7.5 while Figure 2 is pH 7.6. Which one is correct?

pH 7.6 is correct. Fixed

4. Table S3: “Protonation states for critical residues are highlighted in red.” No residue was highlighted.

Fixed

Reviewer #2 (Remarks to the Author):

This paper by Gari et al. describes the use of combined high-speed Atomic Force HS-AFM-HS, single channel electrophysiology and molecular dynamics simulations to monitor conformational dynamics in OmpG. Gari et al. correlate structural changes observed using HS-AFM-HS to channel dynamics from electrophysiology measurements.

HS-AFM imaging of OmpG in lipid bilayers at 10 frames per second: This section describes the structural changes in OmpG observed using HS-AFM. These measurements correlate with those in the literature. There main points as I read this section are; that the substructure of OmpG is visible by HS-AFM including the protrusions of loop 2 and 6 are visible by HS-AFM, that the flexibility of loop 6 – the key mobile & functional structure of this pore can be mapped by HS-AFM, and that there appears to be a visible difference in the dimeric OmpG structures between the open and closed states. I have a few questions to clarify the conclusions being drawn from these data, and would I believe a number of modifications to Figure 1 would make direct comparisons were drawn between the two structures simpler to see.

How does the outline/correlation average for the closed state compare with that of the open state?

We have added dashed outlines of the open state (1b,c) in the closed state average (1h). While the open state structural assignment was rather straightforward, the closed state structural alignment is based on the lattice lines (molecular packing) in the 2D-crystal, based on the open state assignment (note the zig-zag packing of the dimers in the dimer-rows).

Why is the height so much lower in the closed state? Is this an artefact of the low pH?

We provide in the captions of figure 1b and 1g, the values (full color scale: 0.0nm < height < 1.25nm) for the open state and (full color scale: 0.0nm < height < 0.7nm) for the closed state, where the membrane level was set to 0.0nm. The height difference of ~0.5nm corresponds well to the protrusion of Loop-6 in the open state. This is similar to the protrusion difference of loop-6 in the X-ray structure, and in agreement with the Z-directional analysis of the loop motions in MD simulation in Fig.3c. In our revision, we clarify this height difference (line 121).

Could clearer comparisons between standard deviation measurements of loops and correlation averaged structures be shown? This could include rotating these to be at the same angle to facilitate easier comparison. Interesting observations include the potential stabilization of loop 6 in the centre of the dimer for the closed state, and potential increase of mobility in other loops e.g 2 in the closed state?

We preferred to add improved outlines to figure panels 1c, d and 1h, i. These should facilitate feature recognition of the structural elements, while the average and standard-deviation maps are aligned in agreement to the respective raw-data panels (and the supplementary movie). We agree that the SD map indicates higher mobility in the region of Loop-2 in the closed state. Loop-4 also appears more mobile at acidic pH (line 128).

Figure 1 minor observations:

Arrowhead in t=11.8s: loop-6 fluctuating over the lumen. Arrowhead in t=12.0s: Fully open state. I can't see the arrowheads described.

We apologize—the arrowheads must have disappeared during PDF conversion. They should be visible in the revised versions and are described in the caption.

c) Correlation average of OmpG dimers. How was this outline generated?

The outline shown in 1c is based on the molecular structure shown in 1e, now detailed in the caption.

h) Correlation average of OmpG dimers. For comparison, the topography outline of the open state (c) is shown (the position of loop-6 is indicated by the asterisk) I find this confusing – the overlay in c is of the same data set, and here it is from the dataset in c but rotated? This needs to be made clearer or changed to show the topography outline of the open state.

All outlines in figure 1c, d, e are in the same orientation. Since the lattice of the closed state imaging analysis is different from that of the open state (we added dashed lines in figure 1a and 1f to highlight the dimer row orientation), the outlines in 1h,i are accordingly rotated, while the closed-state structure in figure 1j is shown in the same orientation as the structure in 1e.

i) Standard deviation (std) map (n=2472) from the averaging process in (c) Is this (g)? Loop-2 (L2) does not form a β -strand in the closed state, in agreement... In agreement with what?

The Reviewer is right—this should be (g); thank you. This typo has been corrected. Loop-2 does not form a β -strand in the closed state in agreement with the absence of topography pointing away from the barrel in the closed state (h) as compared to the detection of topography in this location in the open state (c). We have amended the caption for clarity.

Correlation between ion flux and loop-6 dynamics of OmpG in lipid bilayers

The single-channel recordings carried out showed good agreement with the literature. Gari et al. observe dwell-times in the millisecond time range, and use HS-AFM-HS to increase the time resolution of their AFM setup to capture these transitions. Gari et al. use the tip position as an indicator of the pore state (open vs closed), with a height change of 5Å observed between the two. Gari et al. report a difference in kinetics as observed by HS-AFM-HS, vs single-channel recordings. The open state time constant is 5x shorter in HS-AFM-HS measurements, and the rest are 2-4 times shorter. The authors attribute this to that loop-6 structural transitions occurring faster than the corresponding functional state transitions in electrophysiology. However the authors do not mention the impact of the tip here. Could the tip impact this transition? How do the authors address the presence of the tip, both in terms of local charge, flow and fluid dynamics?

The oscillation frequency of the probe is ~500-600 kHz in water (~1.5μs per cycle). At this oscillation frequency, the time during which the probe contacts the sample is ~200ns, i.e., ~10% of the oscillation cycle. (Lin, Nature 2019). Thus, the tip interaction is orders of magnitude shorter than the conformational dwell-times, we are, thus, not looking at the tip pushing the loop down or keeping it out at the relevant time scales. Moreover, the oscillation amplitude is ~3nm, thus lifting during each oscillation cycle far above Loop-6, irrespective of its current state. Our AFM probe is made of high-density carbon (HDL). We, therefore, do not expect significant charge effects. In addition, charge effects of the tip should be shielded by the electrolyte in the solution (150mM KCl). With regard to hydrodynamic pressure, we have to consider the motion of the sample stage and the cantilever. The sample stage moves by only ~0.5nm, while the cantilever oscillates with a ~3nm amplitude, which should dominate the relative flow dynamics. In this respect, the tip is ~2nm in diameter, and, more importantly, has a height of ~2.5μm to avoid hydrodynamic pressure on the sample by the oscillating cantilever (Annu. Rev. Biophys. 2013. 42:393–414). In short, we do not think that oscillations, tip charge, and fluid flow would influence the kinetics, but admit that these considerations are justified and ought to be mentioned and discussed. We have accordingly added a paragraph in the methods section (lines 441-457).

In Figure 2, the closed states appear bimodal, and in the HS-AFM-HS, these second peaks are no longer visible, could this be due to the impact of the tip, or due to the increase in noise of the HS-AFM-HS measurements?

The Referee is right—there is a bimodal distribution of the closed state in electrophysiology, while we only (mainly) detect one component in HS-AFM-HS. In HS-AFM-HS, we are only sensitive to Loop-6 motions, while electrophysiology measures also other events that may interrupt the ionic current (flickers).

It is true that increased instrumental noise could contribute to the shorter dwells if they were interpreted as state changes by the StaSI algorithm. We cannot entirely exclude this, but think that it was mainly Loop-6 fluctuations to intermediate coverage of the pore that ‘cut’ the longer dwells into shorter dwells. In our revision, we have added some text about the differences in the dwell-time distributions (lines 154-156, 195-200).

Overall as the next section shows, some of the HS-AFM-HS measurements could show ‘closed’ states for those which are partially open, and I would encourage them to clarify/moderate their structural observations by HS-AFM-HS (e.g, the use of apparent in the next section and compare these to other measurements. e.g. NMR data to gauge the likelihood of these AFM data revealing the ‘true’ conformation.

Paramagnetic relaxation NMR measurements showed that Loop-6 and Loop-2 moved between different conformations in a correlated fashion (Zhuang et al). In another study, (Zhuang et al, Angew. Chem. Int. Ed. 2014, 53, 5897–5902), tethering Loop-6 by dodecylation to the bilayer permanently opened the channel. However, other loops were shown to contribute to channel gating to various degrees and in cooperation with Loop 6. NMR showed that neighboring Loop-5 and Loop-7 were affected by the pinning of Loop-6. In our revision, we discuss further the dynamics reported by NMR and how the HS-AFM-HS measurements relate to structures (lines 203-206).

Analysis of OmpG gating from atomistic simulations

Gari et al. then use MD simulations to gain atomistic structural insights into the dynamics and interactions of loops in OmpG. This section provides structural information to explain why the HS-AFM-HS disagrees with the electrophysiology measurements. This section expands on the flexibility of loop 6 observed in Figure 1 using HS-AFM, to provide atomistic detail of partially closed structures, explaining the differences in structural/functional studies using electrophysiology and HS-AFM-HS in Figure 2. Can the authors reflect on this, linking flexibility to conductance and linking observations in Fig 1d/I to the MD simulations?

In the MDS we observe a highly mobile Loop-6 at neutral pH, in stark contrast with the restrained conformation characteristic of the same loop at acidic pH. In the standard deviation map (1d, pH 7.6) Loop-6 fluctuations are located on both sides of the β -barrel, in agreement with an extended average Loop-6 position with large wavering amplitude on both sides of the barrel. The standard deviation map (and of course the average topography) of the closed state (1i, pH 5.0) shows that Loop-6 is located mainly over the barrel only. This appears in good agreement with the MDS, where at pH 7.5, the loop is found at an average z-axis (δ value) position of $\sim 27\text{\AA}$ with a standard deviation of $\sim 1.9\text{\AA}$, while at pH 5, the loop is found at an average z-axis (δ value) position of $\sim 21\text{\AA}$ with a standard deviation of $\sim 0.9\text{\AA}$. Thus, qualitatively speaking, both techniques agree on the different location and amplitude of motion of Loop-6, in one pH condition, far above the barrel, and in the other pH condition, close to the barrel opening (lines 265-268).

Some of the figure references in this paragraph are hard to follow – could the authors check through (e.g. compare Figure 1e with Figure 1a and single reference to Figure 1e) If these are correct could the authors provide more information as to what is being compared/shown?

We apologize that the referencing to the figure panels was incorrect in this paragraph. In our revision of the manuscript, we now discuss the qualitative agreement of neutral pH loop MDS fluctuations (3b,d, blue traces with HS-AFM standard deviation map (1d)) and acidic pH loop MDS fluctuations (3b,d, red traces with HS-AFM std map (1i)). We corrected the references to the electrophysiology panels, which were simply wrong (now 2e, and 2a).

Free-energy landscape of OmpG gating

Gari et al. then bring together their AFM measurements and MD simulations to define the free energy landscape of OmpG gating, with the main finding being that a tightly closed conformation, formed through a cooperative partnership between loops-1 and -6 works to block the porin. The authors do not reference the position of loop 1 in their HS-AFM measurements. I would be interested to know if they can observe, or calculate the position of loop one in their HS-AFM measurements, and see any local stabilization of loop 6 via images, or standard deviation maps.

Most unfortunately, no—our current HS-AFM imaging and HS-AFM-HS data do not allow us to make statements regarding the position and motion of Loop-1. In our revision, we have integrated this point in the discussion (lines 363-366).

The discussion of the paper is balanced, and reflects on the use of the three techniques to access currently hard to carry out complete biophysical characterisation of molecular structures. The statement regarding discrepancies between HS-AFM-HS and electrophysiology below could be modified to address the presence of the tip in HS-AFM-HS and its potential to lower the barrier.

“The free-energy differences between the states from electrophysiology and height-spectroscopy are similar at both pH, though the results from electrophysiology would, based on the slower transition kinetics, suggest higher barriers than height-spectroscopy.”

The Referee is right. As discussed above, while we do not think that the tip pushes Loop-6 into or prevents Loop-6 from accessing certain positions, it is reasonable to assume that HS-AFM-HS provides thermal energy to the system that lowers the barrier between states. In our revision, we have integrated this point in the discussion (lines 351-353).

Finally – as outlined above – I would be interested to see if the authors see any evidence of this tightly closed state below through HS-AFM imaging or standard deviation measurements as discussed in Figure 1.

No, unfortunately, our data do not provide such an information. It would be interesting to make a Loop-1 deletion and see whether particularly long closed states disappear in electrophysiology traces? This could be the object of a future study. In our revision, we have integrated this point in the discussion (lines 363-366).

“Another interesting finding of the MDS is the tightly closed state, in which, in addition to loop-6, loop-1 is oriented towards the β -barrel lumen, where they interact with each other. Future investigations might resolve kinetic features, conformationally and functionally, that may correspond to this tightly closed state.”

We agree with the Reviewer. We think that Loop-6 is the major player, which physically closes the barrel, but the interplay of Loop-6 with other interacting loops should be an interesting avenue to explore. So far, we know from paramagnetic relaxation NMR measurements that Loop-6 and Loop-2 can move in a correlated fashion (Zhuang et al, JACS 135, 1501). In another study, pinning Loop-6 influenced neighboring loops (Zhuang et al, Angew. Chem. Int. Ed. 2014, 53, 5897–5902). Our MDS suggest, indeed, that Loop-1 could anchor Loop-6 in a tightly closed state, which could be experimentally tested using deletion mutants.

Reviewer #3 (Remarks to the Author):

The authors of this well written paper employed high-speed atomic force microscopy height spectroscopy (HS-AFM-HS) and electrophysiology to characterize the open-closed gating of the outer membrane protein G (OmpG) porin of E. coli. The experimental results are complemented with atomistic molecular dynamics simulations. The overall goal was to explain the previously well-established pH-dependent gating of this porin by structural measurements of the loop dynamics of OmpG, particularly loop 6, which is known to play a major role in opening and closing the pore of this beta-barrel protein.

HS-AFM-HS is an excellent technique to follow the motions of loop 6 in the gating process with adequate time and spatial resolution. The presented experiments are of high technical quality and demonstrate in this well-chosen model system that the technique is suitable to interpret the electrophysiological data by monitoring height fluctuation on an Angstrom length scale with 10-50 microsecond time resolution. The height fluctuations adequately interpreted as loop 6 motions nicely correlate with the electrophysiological opening and closing dynamics at neutral and acidic pH showing motions in the 100 microsecond to 2 millisecond range. Although the MD simulations qualitatively support the experimental results, the technical limitation to 3 microsecond total simulation times limit a direct quantitative comparison with the experimental data on the same time scale.

The authors are encouraged to consider the following points in a revision:

1. Loop 2 is seen in the averaged AFM contours in Fig 1c (neutral pH), but not in Fig 1h (acidic pH). Perhaps this is real and the crystal structure contour in Fig 1j is due to crystallization conditions. Commenting on the choice of these contours will be helpful.

The contour is based on the open OmpG structure (Fig 1e), and was maintained in Fig 1j to highlight the differences, such as the Loop-2 displacement. The contour outlines only serve as a visual guide to locate Loop-6 and Loop-2 in the topography (or the absence thereof in the closed state). Now included in the figure 1 caption.

Below, we have analyzed the crystal packing of structures 2IWW and 2I WV in Yildiz et al. EMBO J 2006. From this analysis it appears that Loop-2 might be involved in the packing in the open state, but not in the closed state. While we could not find any detailed information in that paper, the authors comment: "Otherwise the hydrophobic interactions between OmpG monomers in all six crystal forms we obtained appear to be different, which in itself is a strong argument against an oligomer." In this case, the authors make an argument about the oligomeric state of OmpG, albeit it appears that they had six crystal forms with different packings, and, thus, one would expect them to have commented as to whether Loop-2 was significantly different from one packing to another.

closed state 3D crystal packing (PDB 2IWW)

open state 3D crystal packing (PDB 2IWW)

2. Zhuang et al. (JACS 135, 1501) conducted a comprehensive study of the structural dynamics of OmpG by NMR, related it to electrophysiological data like in the current work, and found by paramagnetic relaxation enhancement measurements and ensemble averaging that loop 6 and to a lesser degree other loops (e.g. loop 2) moved between different conformations in a correlated fashion. It would be interesting to compare this work with the current work in the Discussion and also mention this in the Introduction when discussing that x-ray structures were able to distinguish between open and closed forms of OmpG.

We thank the Referee for the suggestions. In our revision, we have included this work in the introduction and discussion. We cannot conclude about the Loop-2 motions in our HS-AFM-HS data, because we cannot disentangle the height fluctuations and the HS-AFM-HS data should be dominated by Loop-6, which extends the farthest. On the other hand, the fact that we see different Loop-2 average configurations from HS-AFM imaging might be an indirect and static signature of the correlated Loop-6/Loop-2 action, as observed by NMR (lines:65-69, 129-132).

3. The MD simulations were run for a total of 3 microseconds and the simulation analysis of OmpG gating is based on residue R228 (loop 6) motions in this time window. However, NMR relaxation dispersion results of loop 6 residues in Zhuang et al (Supp Fig 1) show that these residues move on a ~500 microsecond timescale (reassuringly very close to the time scale observed by HS-AFM-HS!). Therefore, loop 6 may undergo larger excursions than sampled in the current work by MD. This should be acknowledged. Larger and slower motions of this (and other smaller) loop(s) would still be consistent with the observed HS-AFM-HS data, but their interpretation should not be constrained to the 3 microsecond window that was accessed by the current MD simulations. – Even so, the simulations are certainly valid and help with the interpretation of the paper, but perhaps in a more qualitative than in a detailed quantitative fashion.

We thank the Referee for the very thoughtful comment on correlating the NMR relaxation times of Loop-6 with our HS-AFM-HS data. In our revision, we discuss this point in detail, and have toned down the more quantitative correlations between the MDS fluctuations with our electrophysiology and HS-AFM-HS measurements. We state, however, that appreciably longer simulations would be needed to get insights into full Loop-6 gating, and our major focus insofar as MDS are concerned is characterization of the transitions by means of PMF calculations (lines 203-206).

Minor:

4. Lipid:protein ratios are said to be 0.5-0.7 in the AFM samples. It may be more informative to give those as molar rather than weight ratios.

We usually think in (w:w) for our reconstitution experiments (this gives us a rough and intuitive estimate of the 2D area occupied by both components), but in revision we added molar description 0.65 mM lipid/0.03 mM protein to 0.92 mM lipid/0.03 mM protein (line 92).

5. Under these L:P conditions, are most proteins observed in crystalline arrays by AFM and do most observed sample areas show crystals? Are there also many areas with non-crystalline proteins or bare membrane areas?

At such low LPRs most—if not all sample areas show crystals, or very densely packed vesicles. Please, see below a representative negative stain EM image of a membrane reconstituted OmpG 2D-crystal at such low LPR.

6. The nomenclature of PsubL and PsubO on p. 5 is not clear.

We have added a sentence in the text on page 5 (lines: 178-179, 184) “ P_L represents the open probability from height spectroscopy, where L is the low-height state (i.e., open state)”. “ P_O represents the open probability”

7. In Fig 2a,b,e,f, please explain the red idealized traces that track the gray raw data.

Now included in the figure caption.

8. Fig 2e shows height transitions at pH 7.6 of about 5Å, but they are smaller (~2Å) in Fig 2f at pH 5, contrary to the statement on p5, 1.147.

Yes, height differences in the height-time trace at pH 5 is smaller (Fig 2f and figS4c). Please, see below a filtered version of the trace. There are two main reasons that influence the height difference between states in these traces: First, the tip radius: A blunter tip will not enter the pore significantly, thereby diminishing the full amplitude of the depth signal. Second, the precise position of the tip on the molecule: Only at very precise locations on the protein the full amplitude of motion is recorded. As soon as the tip drifts a bit away from that location, the amplitude of motion shrinks (see Matin et al. Nature Comm 2020). We are primarily interested in the kinetics of state changes, the overall heights being better analyzed in imaging.

9. Fig 4: “physiological” should be called “neutral” pH. It is not clear what physiological means for the outer environment of E. coli.

We thank the Reviewer for pointing this out. We have made the necessary changes throughout the manuscript.

10. Methods, height spectroscopy: if I understand correctly, the cantilever xy scan is stopped at some point after imaging over a crystalline array before switching to height mode and this stop would have to occur over the central hole of the barrel for the method to work. So, what is the success rate of stopping in a successful place over a single OmpG protein vs any other random place? – Or, is this always successful as long as you are over a crystalline array?

In response, we provide more details in the methods section. Before entering height spectroscopy mode, we first we collect images typically at very small scan range of 30 x 30 nm and bring one OmpG molecule to the center of the image, and then turn off the xy piezos. The crystalline array is advantageous because instrumental drift will place the tip sooner or later on another molecule – placement imprecisions and stage drift is also the reason why not always the full amplitude of motion is recorded (see response above). However, we typically collect traces for 10 to 30 s right after targeting a molecule and then switch back to imaging to make sure a molecule is at the center and re-center. The success rate is likely around 50% for the initial second after targeting (lines: 430-439).

Reviewers' Comments:

Reviewer #1:

Remarks to the Author:

In the revised manuscript, the authors have meticulously addressed the raised issues. I am satisfied with the quality of the work and recommend it for publication in Nature Communication.

Reviewer #2:

Remarks to the Author:

The authors have addressed my comments and I recommend publication of this manuscript.

Alice Pyne

Reviewer #3:

Remarks to the Author:

The authors have done a fine job responding constructively to all of my and I believe also the other reviewers' questions. As far as I am concerned, I recommend acceptance in the current form.

REVIEWER COMMENTS

Reviewer #1 (Remarks to the Author):

In the work by Gari R. et al. showed the use of high speed-atomic force microscopy-height spectroscopy (HS-AFM-HS) to study the structural dynamics of an outer membrane protein G (OmpG) channel. In HS-AFM-HS, the AFM tip placed at fixed x-y coordinates oscillates at the z-axis with an amplitude of 3 nm. This approach allows the data (height at the z-axis) to be acquired at a maximal sampling rate of 0.5 us, compared with the AFM imaging rate of 100 ms. OmpG is a membrane channel that shows a pH-dependent gating, the mechanism of which was well studied by the single-channel electrophysiological approach. By fixing the AFM tip's x-y position, the HS-AFM-HS measures the height variation induced by the structural changes of the flexible loops of a single OmpG channel.

While the HS-AFM-HS approach could potentially be used for directly observing the structural dynamics of a membrane protein channel at the high-temporal resolution, there is a lack of discussion about what additional information about the OmpG gating has been revealed by HS-AFM-HF that was not previously learned by single-channel recording. In addition, because the HS-AFM-HS is essentially a single-molecule approach, the experiments should include more rigorous statistical analysis about the single-channel variation. The detailed comments are listed below:

We thank the reviewer for the thorough assessment of our work. As the Reviewer points it out, we used OmpG as a model system to explore the possibilities of HS-AFM-HS for the analysis of membrane protein structural dynamics.

We regret that we have not been able to emphasize sufficiently what additional or different (complementary) information can be gained from HS-AFM-HS as compared to electrophysiology. In a nutshell, the electrophysiology measurements report about ion flow through the channel—or not, i.e. gating. However, the information about the ion flow does not necessarily relate directly to the conformational dynamics. It was our objective to provide with HS-AFM-HS a tool to measure conformational 'gating' at similar temporal resolution as electrophysiology (lines 321-322).

Our results show that OmpG undergoes faster conformational fluctuations as compared to single channel recordings. MD simulations on the microsecond timescale echo the HS-AFM-HS results. HS-AFM-HS captures loop-6 fluctuations in varying positions over the β -barrel, in a series of configurations where the ion conductive pore is not necessarily closed to ion passage. Thus, the measured structural dynamics exceed the functional gating dynamics.

Regarding the single-molecule analysis: We have analyzed HS-AFM-HS data carefully using the STaSI algorithm, which provides unbiased step-transition detection and hierarchical segment clustering. As a result, the algorithm proposes an optimum number of states, weighing between the complexity of the model and the goodness of the fit ('minimum description length', MDL). We further enriched our supplementary table 2, in which we now report about the individuality of the OmpG molecules analyzed.

1. Figure 1. the interpretation of the HS-AFM imaging of OmpG is debatable. It seems to this reviewer another possibility to interpret the topography is to fit the highest region with loop 3 and 4 as the recent NMR structure of OmpG in lipid bilayer showed an extended barrel structure at loop 3 and loop 4 while other regions had a shorter barrel (Retel JS et al. 2017, <https://doi.org/10.1038/s41467-017-02228-2>).

The Referee brings up a valid point regarding our assignment of the AFM topography (Figure 1) with respect to the molecular structure. As the Referee can see in our figure 1b, in AFM, the OmpG barrel (open state) has one side that displays high protrusion, and one side that displays low protrusions. A close look at the X-ray structure (open state) reveals low protruding features on one side Loop-7, Loop-1, Loop-2 (see sideview). Thus, this global assessment basically constrains the barrel placement to one position. In

addition, this placement sets Loop-2 precisely at the location where AFM detects a laterally protruding 'nose'. Loop-2 forms a continuation of the β -strands in the X-ray structure, which is a sturdy secondary structure element that we would expect to give a good signal in AFM. Finally, this placement, places Loop-6 to the highest protrusion, which in addition is the region that showed largest mobility in the SD-map. Thus, based on these four evidences, there was literally only one reasonable way to place the molecular structure into our topography.

Left: Sideview on OmpG: Loop7, facing right, Loop-1 and Loop-2 facing the viewer are protruding much less than loops on the other loops.

Right: Topview on OmpG: Loop-2 forms two β -strands that laterally protrudes from the barrel circumference.

These features, together with Loop-6 being the most protruding, served to assign the HS-AFM average topography to the molecular structure, giving only one reasonable placement.

Our assignment of the closed structure is almost entirely based on the lattice lines as defined by the open structure.

We think, however, that the Referee makes a justified point, regarding the NMR structures in Retel et al. We note that Loop-4 is still in the high protrusion area of our topography. Maybe, Loop-3 is too floppy to give a strong topographical signal in AFM imaging.

In response, we have added a clarifying paragraph that our assignment is just the most likely structural correspondence. We also refer to the NMR structure paper and discuss the location of Loop-3 and Loop-4. We note that the precise structural alignment is not of significance for the remainder of the manuscript (lines 104-115)

2. HS-AFM-height spectroscopy recordings: it is not clear how the x-y coordination of the probe was decided. Was the probe placed at the center of the OmpG channel? What is the resolution of the probe at the x-y surface? How does the position of the probe affect the recording? For example, could the height spectroscopy trace vary with different locations of the tip: a tip positioned at the center of the channel versus one closer L6?

The positioning of the HS-AFM probe when engaging into HS-AFM-HS is at the center x-y coordinates ($x/2$, $y/2$) of an image with dimensions x , y . Before engaging into HS-AFM-HS, we recorded images at small scan size, 30×30 nm, thus centering a single OmpG molecule where the HS-AFM probe is placed.

It is difficult to define the resolution of the probe. However, as the Referee can see based on the images (Figure 1), the probe can resolve details on the structure of the OmpG channel. Conceptually, the probe resolution is not such an important topic for HS-AFM-HS. One can imagine that a blunt hemisphere will still report height changes, as long as the topic of interest is the most protruding structural element (which we believe is the case here), and the tip can still penetrate to some extent into the pore. Thus, the tip radius will ultimately influence the power of the amplitudes recorded, but should not influence the kinetics, which is our main objective here. Based on our imaging, we tested that all probes used for HS-AFM-HS could

resolve OmpG as a pore-like molecule, and, therefore, give the strongest topographical signal between an open and a plugged pore.

Of course, there is lateral stage drift. Thus, as a function of time, the probe will experience many places on the OmpG 2D-crystal, even those of no interest, e.g., between molecules. At these locations, there are essentially no height changes detected, until the lateral drift brings the tip back on top of a channel. Since the height difference is most important for Loop-6 plugging the central pore—or not ($>5\text{\AA}$) we only analyzed periods in our HS-AFM-HS height-time traces that displayed conformational fluctuations larger than $\sim 2.5\text{\AA}$ to eliminate nonspecific height changes. In order to minimize drift issues, we typically collect height spectroscopy data for 10 to 30 seconds and go back to imaging mode, to replace the tip to the center of mass of an OmpG channel.

We have added in the materials and methods section some methodological details regarding the placement of the HS-AFM probe, and associated experimental details with respect to the cantilever and tip properties (lines 431-457).

3. What is the oscillation frequency of the probe? Would the tip affect the dynamics of a flexible polymer? One can imagine when the tip is inside of the barrel, it will prevent L6 from evading into the lumen; therefore the tip could preferentially select an open state. How does the oscillation frequency of the tip affect the height spectroscopy trace of the OmpG pore?

The oscillation frequency of the probe varies between 500 and 600 kHz in buffer solution. At this oscillation frequency, the oscillation cycle is $\sim 2\ \mu\text{s}$. However, the time during which the probe is in contact with the sample is only $\sim 10\%$ of the oscillation cycle, when the cantilever is in its lower swing, thus only $\sim 200\ \text{ns}$ or less. Importantly, the oscillation amplitude is larger than the height of Loop-6, and, thus, during every oscillation cycle, the tip moves far above the space that the loop explores. We, therefore, think that the HS-AFM probe does not prevent L6 from evading the lumen.

HS-AFM only works with Ultra-Short Cantilevers (USC), $8\ \mu\text{m}$ short and high resonance frequency. We cannot test different frequencies, as we have to operate the cantilever at resonance. We reason also—based on the results where the open and closed events are in the $100\ \mu\text{s}$ to $1\ \text{ms}$ time range—that the probing with oscillation frequency of $2\ \mu\text{s}$ should not be a limiting factor.

We added a paragraph detailing the time-scale ($\sim 200\ \text{ns}$) of the probe-OmpG interaction with respect to the characteristic timescales of the channel's conformations. We also comment on the probe oscillation amplitude, because it is important to detail that the probe retracts beyond the height of Loop-6 in every oscillation cycle (lines 442-457).

4. How long can a single membrane protein channel be recorded for? Does the tapping tip have an impact on the structural stability of the membrane protein channel?

In principle, we can image or collect height spectroscopy for hours on a membrane without damaging the sample. However, the time-limiting aspect in HS-AFM-HS is the lateral drift, meaning for how long will the tip stay on the same $\sim 2 \times 2\text{-nm}$ region, the area of a channel pore. We are confident that we can probe the exact same pore for several seconds (see Supplementary Figures), after which, we re-center a channel in imaging mode and switch X and Y scanning off again to record HS-AFM-HS traces (lines 431-439).

5. Figure S4, the authors showed two traces of height spectroscopy recordings at pH 7.5 at different time scales (4s segments vs. 0.4s segments). Are these two traces from the same recording of an OmpG channel or two separated recordings? If the two traces were from the same recording, can the author explain why the histograms of the two traces vary so much? By estimation, the open probability of the Fig S4b trace could be around or smaller than 85%, much less than the 95% of the Figure S4a. Similarly, the two traces at pH 5.0 have shown an even greater difference than those at pH 7.5. The open probability of the Fig. S4c trace could be approximately 10% by estimation.

The Referee is right—the 4s and 0.4s segments of traces are from different OmpG channels at both pHs. We revised supplementary table 2 to further highlight the individuality of the single-molecule experiments. Indeed, two of them had open probabilities of 0.85 and 0.89 at pH 7.6. Open probabilities can vary between channels (and likely with time). Regarding—pH 5.0 in Fig.S4c and Fig.S4d, the trace presented in Fig. S4c was divided into segments of ~0.4s, because of a baseline drift. All open probabilities were calculated from the state assignments by the STaSI algorithm on height spectroscopy traces. However, the algorithm struggled assigning states in extended traces when a baseline drift occurred, because the amplitude of the conformational change is only on the order of 5Å, and the baseline drift can be of such a magnitude. The trace in Fig.S4c contains a baseline drift of over 4s, and the overall height histogram, therefore, convolves the smaller amplitude excursions to the open state, as are better visible when histograms from short segments are visualized as in Fig.S4d.

In our revision of the manuscript, we provide in Supplementary Table 2 additional detail in the individuality of the channels, and describe the need to divide long traces into segments, because fluctuations in the baseline created problems to the analysis software (Lines 463-465).

6. Figure 2c: the dwell time histograms of the closed states at pH 7.6 and 5.0 showed two well-separated peaks indicating two types of gating events at different time scales. However, only a single time constant was shown for both cases. Please include the error for the time constant calculation. Also, please clarify which population of the event was used to derive the time constant.

The Referee is right—there are two types of gating events in the dwell-time distribution in 2c (also in 2d). As we mentioned it in the text, we associate the longer dwells (slower gating events) as conformationally relevant events with respect to fluctuations of the entire Loop-6, while we name the shorter events (100µs and shorter than our bandwidth) ‘flickers’ (i.e., short interruptions of the ionic current possibly due to side-chain fluctuations in the barrel). In our analysis, we report about the longer gating component.

In our revision, we provide errors for each time constant (lines 148-151), and state in caption of figure 4 that the rate constants refer to the slower gating events with respect to Loop-6 motions for both techniques, and not to the short flickers detected in electrophysiology (commented in lines 154-156).

7. Figure 2: HS-AFM height spectroscopy recordings of OmpG. At pH 7.6, the height of the open state was around 0.5 nm and the closed state was above 1.0 nm. In contrast, at pH 5.0, the height of the open state was closed to 0.2 nm and the open state was at the 0.5 nm level. Why is there such a large difference in the height of the open and closed states at the two different pH conditions? How was the height of the tip pre-calibrated? Again, in Figure S4, the height levels of the open and closed states at pH 7.5 and pH 5.0 are also different. Moreover, the relative height levels of the open and closed states at pH 7.5 and pH 5.0 are also inconsistent. At pH 7.0, the height changes by about 1.0 nm from the open to the closed state. At pH 5.0, the height changed by 0.5 nm for one channel while the other channel showed a 0.3 nm shift in height. This raises a concern about the criteria to assign the height levels of open and closed states for a recorded trace without referencing to prior knowledge (or when no prior knowledge is available). If the height level 1.0 nm of Figure S4a trace was assigned as the open state, why the same height level of the Figure S4c trace was assigned as the closed state?

The Referee makes a good point. These height values are not absolute, but relative. In reality, as shown in Supplementary Figure 1, the mica level should be at 0 nm, the surface of the membrane at ~3.7nm, and the uppermost surface of OmpG at ~4.8nm. Thus, the Y-axis values in the HS-AFM-HS traces are merely relative height values with respect to the structural element where the tip landed upon engaging HS-AFM-HS mode. Beside the placement, the height difference between states also strongly depend on the tip sharpness, as the sharper tip can probe deeper inside the tight β -barrel, while the blunter tip cannot.

In our revision, we provide additional experimental details regarding the height values in HS-AFM-HS, and indicate that the tip radius ultimately limits the full amplitude that can be measured in these experiments (lines 436-439).

8. Line 160: the time constants of the open and closed states have no errors. How many OmpG channels were recorded, and what are the variances from channel to channel?

In our revision, we have added errors to the time constants (lines 188-190). In HS-AFM-HS, we have recorded five channels at pH 7.6 and three channels at pH 5.0, and as we mentioned above, we had to segment the individual channel traces on account of a baseline drift (lines 463-465). Further information is now provided in Supplementary Table 2.

9. Page 11 line 302. Different lipid conditions may alter the kinetics. The effect of charged lipids on the OmpG gating was studied by the work below, please cite and discuss the result. (William L. Hwang, Min Chen, Bríd Cronin, Matthew A. Holden, and Hagan Bayley, Asymmetric Droplet Interface Bilayers, Journal of the American Chemical Society 2008 130 (18), 5878-5879, DOI: 10.1021/ja802089s)

We thank the Reviewer for bringing to our attention this question and a paper that shows the effect of ~~the~~ charged lipids on the gating behavior of OmpG. We incorporated this information in our discussion and cited the paper. One would hypothesize that charged lipids interact with the charges borne by Loop-6 (lines 342-348).

10. If the primary goal of the work is to demonstrate the use of the HS-AFM-HS technique for studying the structural dynamics of a membrane channel, please include detailed discussions about the advancement and the limitation of the HS-AFM-HS, particularly as a comparison to the single-channel electrophysiological approach.

We thank the Referee for this suggestion, and we have added the following in the discussion section:

HS-AFM-HS allows the dynamic biomolecular processes occurring under the tip to be measured, from which the diffusion coefficient, surface concentration and oligomeric states of diffusing molecules can be determined, as well as the conformational dynamics¹⁸. Here, we applied this method to measure conformational signatures of channel gating at up to 100 μ s temporal resolution. Single-channel electrophysiology is a powerful technique to measure functional dynamics of channel proteins. As such, the two approaches are complementary. Indeed, it is difficult correlate the functional states determined by electrophysiology with structural states. Functional studies are often correlated with static average structures from X-ray crystallography and cryo-EM. However, while these structures provide crucial high-resolution information about gating residues and selectivity filters, they do not provide intermediate states or kinetic information. Taking OmpG, for example, the X-ray structures at pH 7.5 and pH 5.0 report static open and closed conformations, respectively, while electrophysiology reports that, at both pHs, both states interchange with different kinetics and overall state probabilities. Here, using HS-AFM-HS, we attempted to bridge this gap and access kinetic and probabilistic information from a conformation angle. HS-AFM-HS has currently several limitations: (i) Instrumental drift makes recordings over long experimental time challenging. Currently, we need to switch between imaging and height spectroscopy modes to assure recording on a specific location on top of a molecule. (ii) The conformational signal must be larger than the thermal noise of the cantilever, which might become limiting at high bandwidth. (iii) The temporal resolution is limited by the feedback bandwidth, currently at ~100kHz in ideal conditions. (iv) The sample

is substrate supported, in contrast to a free-standing bilayer in single channel recordings. Thus, HS-AFM-HS closes an experimental gap for the analysis of fast biomolecular dynamics, but remains improvable with future technological efforts. (lines 367-387; last paragraph in the discussion section).

Minor:

1. Page 11 line 329. E. coli should be italic

Fixed

2. Line 343: “Nacl” should be “NaCl”

Fixed

3. Figure S4: The pH of a and b is shown as 7.5 while Figure 2 is pH 7.6. Which one is correct?

pH 7.6 is correct. Fixed

4. Table S3: “Protonation states for critical residues are highlighted in red.” No residue was highlighted.

Fixed

Reviewer #2 (Remarks to the Author):

This paper by Gari et al. describes the use of combined high-speed Atomic Force HS-AFM-HS, single channel electrophysiology and molecular dynamics simulations to monitor conformational dynamics in OmpG. Gari et al. correlate structural changes observed using HS-AFM-HS to channel dynamics from electrophysiology measurements.

HS-AFM imaging of OmpG in lipid bilayers at 10 frames per second: This section describes the structural changes in OmpG observed using HS-AFM. These measurements correlate with those in the literature. There main points as I read this section are; that the substructure of OmpG is visible by HS-AFM including the protrusions of loop 2 and 6 are visible by HS-AFM, that the flexibility of loop 6 – the key mobile & functional structure of this pore can be mapped by HS-AFM, and that there appears to be a visible difference in the dimeric OmpG structures between the open and closed states. I have a few questions to clarify the conclusions being drawn from these data, and would I believe a number of modifications to Figure 1 would make direct comparisons were drawn between the two structures simpler to see.

How does the outline/correlation average for the closed state compare with that of the open state?

We have added dashed outlines of the open state (1b,c) in the closed state average (1h). While the open state structural assignment was rather straightforward, the closed state structural alignment is based on the lattice lines (molecular packing) in the 2D-crystal, based on the open state assignment (note the zig-zag packing of the dimers in the dimer-rows).

Why is the height so much lower in the closed state? Is this an artefact of the low pH?

We provide in the captions of figure 1b and 1g, the values (full color scale: 0.0nm < height < 1.25nm) for the open state and (full color scale: 0.0nm < height < 0.7nm) for the closed state, where the membrane level was set to 0.0nm. The height difference of ~0.5nm corresponds well to the protrusion of Loop-6 in the open state. This is similar to the protrusion difference of loop-6 in the X-ray structure, and in agreement with the Z-directional analysis of the loop motions in MD simulation in Fig.3c. In our revision, we clarify this height difference (line 121).

Could clearer comparisons between standard deviation measurements of loops and correlation averaged structures be shown? This could include rotating these to be at the same angle to facilitate easier comparison. Interesting observations include the potential stabilization of loop 6 in the centre of the dimer for the closed state, and potential increase of mobility in other loops e.g 2 in the closed state?

We preferred to add improved outlines to figure panels 1c, d and 1h, i. These should facilitate feature recognition of the structural elements, while the average and standard-deviation maps are aligned in agreement to the respective raw-data panels (and the supplementary movie). We agree that the SD map indicates higher mobility in the region of Loop-2 in the closed state. Loop-4 also appears more mobile at acidic pH (line 128).

Figure 1 minor observations:

Arrowhead in t=11.8s: loop-6 fluctuating over the lumen. Arrowhead in t=12.0s: Fully open state. I can't see the arrowheads described.

We apologize—the arrowheads must have disappeared during PDF conversion. They should be visible in the revised versions and are described in the caption.

c) Correlation average of OmpG dimers. How was this outline generated?

The outline shown in 1c is based on the molecular structure shown in 1e, now detailed in the caption.

h) Correlation average of OmpG dimers. For comparison, the topography outline of the open state (c) is shown (the position of loop-6 is indicated by the asterisk) I find this confusing – the overlay in c is of the same data set, and here it is from the dataset in c but rotated? This needs to be made clearer or changed to show the topography outline of the open state.

All outlines in figure 1c, d, e are in the same orientation. Since the lattice of the closed state imaging analysis is different from that of the open state (we added dashed lines in figure 1a and 1f to highlight the dimer row orientation), the outlines in 1h,i are accordingly rotated, while the closed-state structure in figure 1j is shown in the same orientation as the structure in 1e.

i) Standard deviation (std) map (n=2472) from the averaging process in (c) Is this (g)? Loop-2 (L2) does not form a β -strand in the closed state, in agreement... In agreement with what?

The Reviewer is right—this should be (g); thank you. This typo has been corrected. Loop-2 does not form a β -strand in the closed state in agreement with the absence of topography pointing away from the barrel in the closed state (h) as compared to the detection of topography in this location in the open state (c). We have amended the caption for clarity.

Correlation between ion flux and loop-6 dynamics of OmpG in lipid bilayers

The single-channel recordings carried out showed good agreement with the literature. Gari et al. observe dwell-times in the millisecond time range, and use HS-AFM-HS to increase the time resolution of their AFM setup to capture these transitions. Gari et al. use the tip position as an indicator of the pore state (open vs closed), with a height change of 5Å observed between the two. Gari et al. report a difference in kinetics as observed by HS-AFM-HS, vs single-channel recordings. The open state time constant is 5x shorter in HS-AFM-HS measurements, and the rest are 2-4 times shorter. The authors attribute this to that loop-6 structural transitions occurring faster than the corresponding functional state transitions in electrophysiology. However the authors do not mention the impact of the tip here. Could the tip impact this transition? How do the authors address the presence of the tip, both in terms of local charge, flow and fluid dynamics?

The oscillation frequency of the probe is ~500-600 kHz in water (~1.5μs per cycle). At this oscillation frequency, the time during which the probe contacts the sample is ~200ns, i.e., ~10% of the oscillation cycle. (Lin, Nature 2019). Thus, the tip interaction is orders of magnitude shorter than the conformational dwell-times, we are, thus, not looking at the tip pushing the loop down or keeping it out at the relevant time scales. Moreover, the oscillation amplitude is ~3nm, thus lifting during each oscillation cycle far above Loop-6, irrespective of its current state. Our AFM probe is made of high-density carbon (HDL). We, therefore, do not expect significant charge effects. In addition, charge effects of the tip should be shielded by the electrolyte in the solution (150mM KCl). With regard to hydrodynamic pressure, we have to consider the motion of the sample stage and the cantilever. The sample stage moves by only ~0.5nm, while the cantilever oscillates with a ~3nm amplitude, which should dominate the relative flow dynamics. In this respect, the tip is ~2nm in diameter, and, more importantly, has a height of ~2.5μm to avoid hydrodynamic pressure on the sample by the oscillating cantilever (Annu. Rev. Biophys. 2013. 42:393–414). In short, we do not think that oscillations, tip charge, and fluid flow would influence the kinetics, but admit that these considerations are justified and ought to be mentioned and discussed. We have accordingly added a paragraph in the methods section (lines 441-457).

In Figure 2, the closed states appear bimodal, and in the HS-AFM-HS, these second peaks are no longer visible, could this be due to the impact of the tip, or due to the increase in noise of the HS-AFM-HS measurements?

The Referee is right—there is a bimodal distribution of the closed state in electrophysiology, while we only (mainly) detect one component in HS-AFM-HS. In HS-AFM-HS, we are only sensitive to Loop-6 motions, while electrophysiology measures also other events that may interrupt the ionic current (flickers).

It is true that increased instrumental noise could contribute to the shorter dwells if they were interpreted as state changes by the StaSI algorithm. We cannot entirely exclude this, but think that it was mainly Loop-6 fluctuations to intermediate coverage of the pore that ‘cut’ the longer dwells into shorter dwells. In our revision, we have added some text about the differences in the dwell-time distributions (lines 154-156, 195-200).

Overall as the next section shows, some of the HS-AFM-HS measurements could show ‘closed’ states for those which are partially open, and I would encourage them to clarify/moderate their structural observations by HS-AFM-HS (e.g, the use of apparent in the next section and compare these to other measurements. e.g. NMR data to gauge the likelihood of these AFM data revealing the ‘true’ conformation.

Paramagnetic relaxation NMR measurements showed that Loop-6 and Loop-2 moved between different conformations in a correlated fashion (Zhuang et al). In another study, (Zhuang et al, Angew. Chem. Int. Ed. 2014, 53, 5897 –5902), tethering Loop-6 by dodecylation to the bilayer permanently opened the channel. However, other loops were shown to contribute to channel gating to various degrees and in cooperation with Loop 6. NMR showed that neighboring Loop-5 and Loop-7 were affected by the pinning of Loop-6. In our revision, we discuss further the dynamics reported by NMR and how the HS-AFM-HS measurements relate to structures (lines 203-206).

Analysis of OmpG gating from atomistic simulations

Gari et al. then use MD simulations to gain atomistic structural insights into the dynamics and interactions of loops in OmpG. This section provides structural information to explain why the HS-AFM-HS disagrees with the electrophysiology measurements. This section expands on the flexibility of loop 6 observed in Figure 1 using HS-AFM, to provide atomistic detail of partially closed structures, explaining the differences in structural/functional studies using electrophysiology and HS-AFM-HS in Figure 2. Can the authors reflect on this, linking flexibility to conductance and linking observations in Fig 1d/I to the MD simulations?

In the MDS we observe a highly mobile Loop-6 at neutral pH, in stark contrast with the restrained conformation characteristic of the same loop at acidic pH. In the standard deviation map (1d, pH 7.6) Loop-6 fluctuations are located on both sides of the β -barrel, in agreement with an extended average Loop-6 position with large wavering amplitude on both sides of the barrel. The standard deviation map (and of course the average topography) of the closed state (1i, pH 5.0) shows that Loop-6 is located mainly over the barrel only. This appears in good agreement with the MDS, where at pH 7.5, the loop is found at an average z-axis (δ value) position of $\sim 27\text{\AA}$ with a standard deviation of $\sim 1.9\text{\AA}$, while at pH 5, the loop is found at an average z-axis (δ value) position of $\sim 21\text{\AA}$ with a standard deviation of $\sim 0.9\text{\AA}$. Thus, qualitatively speaking, both techniques agree on the different location and amplitude of motion of Loop-6, in one pH condition, far above the barrel, and in the other pH condition, close to the barrel opening (lines 265-268).

Some of the figure references in this paragraph are hard to follow – could the authors check through (e.g. compare Figure 1e with Figure 1a and single reference to Figure 1e) If these are correct could the authors provide more information as to what is being compared/shown?

We apologize that the referencing to the figure panels was incorrect in this paragraph. In our revision of the manuscript, we now discuss the qualitative agreement of neutral pH loop MDS fluctuations (3b,d, blue traces with HS-AFM standard deviation map (1d)) and acidic pH loop MDS fluctuations (3b,d, red traces with HS-AFM std map (1i)). We corrected the references to the electrophysiology panels, which were simply wrong (now 2e, and 2a).

Free-energy landscape of OmpG gating

Gari et al. then bring together their AFM measurements and MD simulations to define the free energy landscape of OmpG gating, with the main finding being that a tightly closed conformation, formed through a cooperative partnership between loops-1 and -6 works to block the porin. The authors do not reference the position of loop 1 in their HS-AFM measurements. I would be interested to know if they can observe, or calculate the position of loop one in their HS-AFM measurements, and see any local stabilization of loop 6 via images, or standard deviation maps.

Most unfortunately, no—our current HS-AFM imaging and HS-AFM-HS data do not allow us to make statements regarding the position and motion of Loop-1. In our revision, we have integrated this point in the discussion (lines 363-366).

The discussion of the paper is balanced, and reflects on the use of the three techniques to access currently hard to carry out complete biophysical characterisation of molecular structures. The statement regarding discrepancies between HS-AFM-HS and electrophysiology below could be modified to address the presence of the tip in HS-AFM-HS and its potential to lower the barrier.

“The free-energy differences between the states from electrophysiology and height-spectroscopy are similar at both pH, though the results from electrophysiology would, based on the slower transition kinetics, suggest higher barriers than height-spectroscopy.”

The Referee is right. As discussed above, while we do not think that the tip pushes Loop-6 into or prevents Loop-6 from accessing certain positions, it is reasonable to assume that HS-AFM-HS provides thermal energy to the system that lowers the barrier between states. In our revision, we have integrated this point in the discussion (lines 351-353).

Finally – as outlined above – I would be interested to see if the authors see any evidence of this tightly closed state below through HS-AFM imaging or standard deviation measurements as discussed in Figure 1.

No, unfortunately, our data do not provide such an information. It would be interesting to make a Loop-1 deletion and see whether particularly long closed states disappear in electrophysiology traces? This could be the object of a future study. In our revision, we have integrated this point in the discussion (lines 363-366).

“Another interesting finding of the MDS is the tightly closed state, in which, in addition to loop-6, loop-1 is oriented towards the β -barrel lumen, where they interact with each other. Future investigations might resolve kinetic features, conformationally and functionally, that may correspond to this tightly closed state.”

We agree with the Reviewer. We think that Loop-6 is the major player, which physically closes the barrel, but the interplay of Loop-6 with other interacting loops should be an interesting avenue to explore. So far, we know from paramagnetic relaxation NMR measurements that Loop-6 and Loop-2 can move in a correlated fashion (Zhuang et al, JACS 135, 1501). In another study, pinning Loop-6 influenced neighboring loops (Zhuang et al, Angew. Chem. Int. Ed. 2014, 53, 5897–5902). Our MDS suggest, indeed, that Loop-1 could anchor Loop-6 in a tightly closed state, which could be experimentally tested using deletion mutants.

Reviewer #3 (Remarks to the Author):

The authors of this well written paper employed high-speed atomic force microscopy height spectroscopy (HS-AFM-HS) and electrophysiology to characterize the open-closed gating of the outer membrane protein G (OmpG) porin of E. coli. The experimental results are complemented with atomistic molecular dynamics simulations. The overall goal was to explain the previously well-established pH-dependent gating of this porin by structural measurements of the loop dynamics of OmpG, particularly loop 6, which is known to play a major role in opening and closing the pore of this beta-barrel protein.

HS-AFM-HS is an excellent technique to follow the motions of loop 6 in the gating process with adequate time and spatial resolution. The presented experiments are of high technical quality and demonstrate in this well-chosen model system that the technique is suitable to interpret the electrophysiological data by monitoring height fluctuation on an Angstrom length scale with 10-50 microsecond time resolution. The height fluctuations adequately interpreted as loop 6 motions nicely correlate with the electrophysiological opening and closing dynamics at neutral and acidic pH showing motions in the 100 microsecond to 2 millisecond range. Although the MD simulations qualitatively support the experimental results, the technical limitation to 3 microsecond total simulation times limit a direct quantitative comparison with the experimental data on the same time scale.

The authors are encouraged to consider the following points in a revision:

1. Loop 2 is seen in the averaged AFM contours in Fig 1c (neutral pH), but not in Fig 1h (acidic pH). Perhaps this is real and the crystal structure contour in Fig 1j is due to crystallization conditions. Commenting on the choice of these contours will be helpful.

The contour is based on the open OmpG structure (Fig 1e), and was maintained in Fig 1j to highlight the differences, such as the Loop-2 displacement. The contour outlines only serve as a visual guide to locate Loop-6 and Loop-2 in the topography (or the absence thereof in the closed state). Now included in the figure 1 caption.

Below, we have analyzed the crystal packing of structures 2IWW and 2I WV in Yildiz et al. EMBO J 2006. From this analysis it appears that Loop-2 might be involved in the packing in the open state, but not in the closed state. While we could not find any detailed information in that paper, the authors comment: "Otherwise the hydrophobic interactions between OmpG monomers in all six crystal forms we obtained appear to be different, which in itself is a strong argument against an oligomer." In this case, the authors make an argument about the oligomeric state of OmpG, albeit it appears that they had six crystal forms with different packings, and, thus, one would expect them to have commented as to whether Loop-2 was significantly different from one packing to another.

closed state 3D crystal packing (PDB 2IWW)

open state 3D crystal packing (PDB 2IWW)

2. Zhuang et al. (JACS 135, 1501) conducted a comprehensive study of the structural dynamics of OmpG by NMR, related it to electrophysiological data like in the current work, and found by paramagnetic relaxation enhancement measurements and ensemble averaging that loop 6 and to a lesser degree other loops (e.g. loop 2) moved between different conformations in a correlated fashion. It would be interesting to compare this work with the current work in the Discussion and also mention this in the Introduction when discussing that x-ray structures were able to distinguish between open and closed forms of OmpG.

We thank the Referee for the suggestions. In our revision, we have included this work in the introduction and discussion. We cannot conclude about the Loop-2 motions in our HS-AFM-HS data, because we cannot disentangle the height fluctuations and the HS-AFM-HS data should be dominated by Loop-6, which extends the farthest. On the other hand, the fact that we see different Loop-2 average configurations from HS-AFM imaging might be an indirect and static signature of the correlated Loop-6/Loop-2 action, as observed by NMR (lines:65-69, 129-132).

3. The MD simulations were run for a total of 3 microseconds and the simulation analysis of OmpG gating is based on residue R228 (loop 6) motions in this time window. However, NMR relaxation dispersion results of loop 6 residues in Zhuang et al (Supp Fig 1) show that these residues move on a ~500 microsecond timescale (reassuringly very close to the time scale observed by HS-AFM-HS!). Therefore, loop 6 may undergo larger excursions than sampled in the current work by MD. This should be acknowledged. Larger and slower motions of this (and other smaller) loop(s) would still be consistent with the observed HS-AFM-HS data, but their interpretation should not be constrained to the 3 microsecond window that was accessed by the current MD simulations. – Even so, the simulations are certainly valid and help with the interpretation of the paper, but perhaps in a more qualitative than in a detailed quantitative fashion.

We thank the Referee for the very thoughtful comment on correlating the NMR relaxation times of Loop-6 with our HS-AFM-HS data. In our revision, we discuss this point in detail, and have toned down the more quantitative correlations between the MDS fluctuations with our electrophysiology and HS-AFM-HS measurements. We state, however, that appreciably longer simulations would be needed to get insights into full Loop-6 gating, and our major focus insofar as MDS are concerned is characterization of the transitions by means of PMF calculations (lines 203-206).

Minor:

4. Lipid:protein ratios are said to be 0.5-0.7 in the AFM samples. It may be more informative to give those as molar rather than weight ratios.

We usually think in (w:w) for our reconstitution experiments (this gives us a rough and intuitive estimate of the 2D area occupied by both components), but in revision we added molar description 0.65 mM lipid/0.03 mM protein to 0.92 mM lipid/0.03 mM protein (line 92).

5. Under these L:P conditions, are most proteins observed in crystalline arrays by AFM and do most observed sample areas show crystals? Are there also many areas with non-crystalline proteins or bare membrane areas?

At such low LPRs most—if not all sample areas show crystals, or very densely packed vesicles. Please, see below a representative negative stain EM image of a membrane reconstituted OmpG 2D-crystal at such low LPR.

6. The nomenclature of PsubL and PsubO on p. 5 is not clear.

We have added a sentence in the text on page 5 (lines: 178-179, 184) “ P_L represents the open probability from height spectroscopy, where L is the low-height state (i.e., open state)”. “ P_O represents the open probability”

7. In Fig 2a,b,e,f, please explain the red idealized traces that track the gray raw data.

Now included in the figure caption.

8. Fig 2e shows height transitions at pH 7.6 of about 5Å, but they are smaller (~2Å) in Fig 2f at pH 5, contrary to the statement on p5, 1.147.

Yes, height differences in the height-time trace at pH 5 is smaller (Fig 2f and figS4c). Please, see below a filtered version of the trace. There are two main reasons that influence the height difference between states in these traces: First, the tip radius: A blunter tip will not enter the pore significantly, thereby diminishing the full amplitude of the depth signal. Second, the precise position of the tip on the molecule: Only at very precise locations on the protein the full amplitude of motion is recorded. As soon as the tip drifts a bit away from that location, the amplitude of motion shrinks (see Matin et al. Nature Comm 2020). We are primarily interested in the kinetics of state changes, the overall heights being better analyzed in imaging.

9. Fig 4: “physiological” should be called “neutral” pH. It is not clear what physiological means for the outer environment of E. coli.

We thank the Reviewer for pointing this out. We have made the necessary changes throughout the manuscript.

10. Methods, height spectroscopy: if I understand correctly, the cantilever xy scan is stopped at some point after imaging over a crystalline array before switching to height mode and this stop would have to occur over the central hole of the barrel for the method to work. So, what is the success rate of stopping in a successful place over a single OmpG protein vs any other random place? – Or, is this always successful as long as you are over a crystalline array?

In response, we provide more details in the methods section. Before entering height spectroscopy mode, we first we collect images typically at very small scan range of 30 x 30 nm and bring one OmpG molecule to the center of the image, and then turn off the xy piezos. The crystalline array is advantageous because instrumental drift will place the tip sooner or later on another molecule – placement imprecisions and stage drift is also the reason why not always the full amplitude of motion is recorded (see response above). However, we typically collect traces for 10 to 30 s right after targeting a molecule and then switch back to imaging to make sure a molecule is at the center and re-center. The success rate is likely around 50% for the initial second after targeting (lines: 430-439).